# Between Stochastic and Adversarial Online Convex Optimization: Improved Regret Bounds via Smoothness

**Sarah Sachs**
University of Amsterdam
Korteweg-de Vries Institute for Mathematics
`s.c.sachs@uva.nl`

**Hédi Hadiji**
University of Amsterdam
Korteweg-de Vries Institute for Mathematics
`hedi.hadiji@gmail.com`

**Tim van Erven**
University of Amsterdam
Korteweg-de Vries Institute for Mathematics
`tim@timvanerven.nl`

**Cristóbal Guzmán**
Pontificia Universidad Católica de Chile
Institute for Mathematical and Computational Eng.
Facultad de Matemáticas y Escuela de Ingeniería
`crguzmanp@mat.uc.cl`

## Abstract

Stochastic and adversarial data are two widely studied settings in online learning. But many optimization tasks are neither i.i.d. nor fully adversarial, which makes it of fundamental interest to get a better theoretical understanding of the world between these extremes. In this work, we establish novel regret bounds for online convex optimization in a setting that interpolates between stochastic i.i.d. and fully adversarial losses. By exploiting smoothness of the expected losses, these bounds replace dependence on the maximum gradient length by the variance of the gradients, which was previously known only for linear losses. In addition, they weaken the i.i.d. assumption by allowing, for example, adversarially poisoned rounds, which were previously considered in the expert and bandit setting. Our results extend this to the online convex optimization framework. In the fully i.i.d. case, our bounds match the rates one would expect from results in stochastic acceleration, and in the fully adversarial case, they gracefully deteriorate to match the minimax regret. We further provide lower bounds showing that our regret upper bounds are tight for all intermediate regimes in terms of the stochastic variance and the adversarial variation of the loss gradients.

## 1 Introduction

Two of the main approaches for solving convex optimization problems under uncertain data are stochastic convex optimization (SCO) [21, 30] and online convex optimization (OCO) [37]. These two models are very different in their assumptions and goals, despite the fact that they share many techniques. In SCO it is assumed that the loss functions follow an independent, identically distributed (i.i.d.) process, and the goal is to minimize the *excess risk*, which is the optimization error under the expected loss. By contrast, in OCO the losses can be chosen adversarially and the goal is to minimize

36th Conference on Neural Information Processing Systems (NeurIPS 2022).

the *cumulative regret*, which is the difference between the cumulative incurred losses over rounds against the best fixed strategy in hindsight. Much less is known about what happens in between, in scenarios that interpolate between the i.i.d. and adversarial settings. This intermediate setting has drawn major attention in the recent years in the expert and bandit setting [14] [2] [36], however, as mentioned in [14], little is known for online convex optimization. Our work studies this in a generalization of the OCO setting, in which nature chooses distributions for the data that may vary arbitrarily over time, and we provide regret bounds in terms of two quantities that measure how adversarial these distributions are. The standard OCO setting corresponds to the case where the distributions are point-masses on adversarial data points.

**Main Contribution.** Our main contribution is a new analysis of *optimistic online algorithms* [25, 26] that takes advantage of smoothness of the expected loss. This analysis allows for a gradual interpolation between worst-case adversarial regret bounds and the best-known expected regret bounds in the stochastic case, and also provides quantifiable improvements for intermediate cases.[1] To capture the full range between i.i.d. and fully adversarial settings, we consider a similar adversarial model as in [27], i.e., nature chooses distributions $\mathcal{D}_t$ in iteration $t$, and the learner suffers loss $f_t(x_t, \xi_t)$ with $\xi_t \sim \mathcal{D}_t$. Importantly, we do not assume that the distributions $\mathcal{D}_t$ are all the same, but they may vary adversarially over time.

To properly quantify the interpolation between the i.i.d. and fully adversarial settings in the regret bound, we introduce two parameters for the loss sequence. Namely the *cumulative variance*, $\bar{\sigma}_T$, which captures the stochastic aspect of the learning task, i.e., the variance of the $\mathcal{D}_t$; and the *cumulative adversarial variation*, $\bar{\Sigma}_T$, which captures the adversarial difficulties of the data, i.e., the difference between $\mathbb{E}_{\xi \sim \mathcal{D}_t}[\nabla f(\,\cdot\,, \xi)]$ and $\mathbb{E}_{\xi \sim \mathcal{D}_{t-1}}[\nabla f(\,\cdot\,, \xi)]$. With these two key quantities, our first main result in Theorem 5 shows that the expected regret $\mathbb{E}[R_T(u)]$, that is the difference between the cumulative losses of the learner and a fixed solution in hindsight, is bounded by

$$\mathbb{E}\left[R_T(u)\right] = O\Big( D(\bar{\sigma}_T + \bar{\Sigma}_T)\sqrt{T} + LD^2 \Big), \tag{1}$$

where $L$ is the smoothness constant of the expected functions $F^t = \mathbb{E}_{\xi \sim \mathcal{D}_t}[f(\,\cdot\,, \xi)]$. If, in addition, the functions $F^t$ are $\mu$-strongly convex, then in Theorem 7 we obtain (below, $\kappa := L/\mu$ is the condition number)

$$\mathbb{E}[R_T(u)] = O\left( \frac{1}{\mu} \left( \sigma_{\max}^2 + \Sigma_{\max}^2 \right) \log T + LD^2 \kappa \log \kappa \right).$$

Both bounds are tight: we prove matching lower bounds in Theorems 6 and 8. In Section 3.1 we show that our results match the known adversarial regret bounds as well as the best results in the i.i.d. case. For the latter, only the linear case was so far obtained directly via a regret analysis (see Sec. 5.2 in [25], and prior work [12, 4]). Using optimistic mirror descent, they obtained the regret guarantee of $R_T(u) \leqslant O(\sqrt{\sum_{t=1}^{T} \|\nabla f(x_t, \xi_t) - m_t\|^2})$, where $m_t$ denotes an optimistic guess of the gradient that is chosen before round $t$. In the i.i.d. case with the prediction $m_t = \nabla f(x_{t-1}, \xi_{t-1})$, this can be shown to imply that the expected regret is upper bounded by $\mathbb{E}[R_T(u)] = O(\sigma\sqrt{T} + \sqrt{\sum_{t=1}^{T} \mathbb{E}[\|\nabla F^t(x_t) - \nabla F^t(x_{t-1})\|^2]})$, where $\sigma$ denotes the variance of the stochastic gradients. This simplifies to

$$\mathbb{E}[R_T(u)] = O(\sigma\sqrt{T}) \qquad \text{for the i.i.d. case with linear functions } \nabla F^t,$$

which is a special case of (1), because $\sigma = \bar{\sigma}_T$ and $\bar{\Sigma}_T = 0$ for i.i.d. losses, and $\nabla F^t(x_t) = \nabla F^t(x_{t-1})$ and $L = 0$ for linear functions. It is not immediately obvious how to generalize this result to general convex functions with smoothness $L > 0$, however. In this case, we can guess the appropriate regret bound based on known convergence results for *stochastic accelerated gradient descent* (SAGD) [17, 8]: if we knew in advance that the losses would be i.i.d. and we did not care about computational efficiency, then we could run a new instance of SAGD for each round $t$. Summing the known rate for SAGD over $t$ then gives $\mathbb{E}[R_T(u)] \leqslant O(\sigma\sqrt{T} + L)$ (for more details see the batch-to-online conversion in Appendix D). This raises the question if a similar bound can also be obtained directly via a regret analysis, without assuming i.i.d. observations in advance. This

---

[1]It is well-known that in the fully adversarial case smoothness does not yield asymptotic improvements on regret [11], whereas for SCO improvements can be obtained only under low-noise [8].

question is then answered by our result ([1]), which indeed reduces to this rate for general convex smooth i.i.d. functions, matching the aforementioned bound up to constants. We achieve this by using smoothness to bound $\mathbb{E}[\|\nabla F^t(x_t) - \nabla F^t(x_{t-1})\|^2] \leqslant L\,\mathbb{E}[\|x_t - x_{t-1}\|^2]$, which can be canceled by a negative quadratic term that we obtain from an improved analysis of the regret. The use of this negative term in the analysis dates back to [20], who used it to achieve an improved $O(1/T)$ rate on the extra-gradient method.

In addition to unifying the analysis of these two extreme cases and obtaining the best-known results via one algorithm (OFTRL ([2]) for convex functions and OFTRL on a surrogate loss ([5]) for strongly convex functions), our results give new insight for intermediate cases. Thus, as a second main contribution we shed light on a setting that is neither fully adversarial nor i.i.d. To illustrate this, we highlight some examples here, which received attention in the recent literature.

**Adversarial corruptions:** Consider i.i.d. functions with adversarial corruptions, as considered in the context of the expert and bandit settings in [14], [2]. If the (cumulative) corruption level is bounded by a constant $C$, in [14] an expected regret bound of $\mathbb{E}[R_T(u)] = O(R_T^s + \sqrt{CR_T^s})$ was obtained, where $R_T^s$ denotes the regret with respect to the uncorrupted data. In [14], the authors raised the question of whether it is possible to obtain regret bounds with a similar square-root dependence on the corruption level $C$ for online convex optimization. Indeed, for this intermediate model, we derive a regret bound

$$\mathbb{E}[R_T(u)] \leqslant O(R_T^s + D\sqrt{GC}),$$

for the general convex case from our Theorem [5]. We elaborate on this in Section [4.2].

**Random order models** The random order model (ROM) dates back to [18] in combinatorial online learning. It has drawn attention in the online convex optimization community as an elegant relaxation of the adversarial model [31, 6]. Complementary to the results in [31], we show that the dependence on $G$ in the regret bound can be reduced to a dependence of $\sigma$, where $\sigma$ denotes the variance of gradients in the uniform distribution over loss functions $f_1, \ldots f_T$. That is,

$$\mathbb{E}[R_T(u)] \leqslant O\left(D\sigma\sqrt{T\log\left(e\frac{\widetilde{\sigma}}{\sigma}\right)}\right),$$

where $\widetilde{\sigma}$ denotes a slightly weaker notion of variance (see Corollary [10]). We derive these results from our main theorem under stronger assumptions than in [31], namely, we require individual convexity of the adversarial choices whereas [31] only assume cumulative convexity. However, we also obtain a better rate with $\sigma$ instead of $G$ as the leading factor, so the results are not directly comparable. We also consider a variant of the random order model, which we call the *multiple pass random order model (multi-pass ROM)*. This is inspired by multiple shuffle SGD and can be considered another intermediate example between adversarial and stochastic data. We elaborate on both examples, i.e., the ROM and multi-pass ROM in Section [4.3].

## 1.1 Related work

As mentioned in the previous section, our work is inspired by results in the *gradual variation* and in the *stochastic approximation* literature. The gradual variation literature dates back to [12], with later extensions by [4] and [25, 26]. In addition to some technical relation to the aforementioned work, there is also a natural relation between our parameters $\bar{\sigma}_T$ and $\bar{\Sigma}_T$ and variational parameters in [33], [12] or [4]. However, as we elaborate in Remark [4], there are some fundamental differences between these variational parameters and $\bar{\sigma}_T, \bar{\Sigma}_T$, which prevent us from directly obtaining a smooth interpolation from these results. It is also interesting to note that there is a relation between $\bar{\Sigma}_T$ and the *path length* parameters considered in *dynamic regret* bounds [35, 34]. However, since their analysis targets a fundamentally different notion of regret, namely dynamic regret, the results are incomparable.

With respect to the results, our findings are fundamentally different from the stochastic approximation literature, since we do not rely on the assumption that the data is following a distribution. However, we were inspired by analysis techniques and the convergence thresholds set by this literature. Our approach of obtaining accelerated rates by negative terms arising from smoothness in a regret bound has previously been used in the context of variational inequalities and saddle-point problems. Using

this idea, [20] obtained improved rates $O(1/T)$ for the extra-gradient method. More recently [17] showed that acceleration in stochastic convex optimization can benefit from negative terms arising in optimistic FTRL via an anytime-online-to-batch conversion [5]. Although an important inspiration for our approach, the techniques of [17] do not directly carry over, because they evaluate gradients at the time-average of the algorithm's iterates, making them much more stable than the last iterate, which comes up when controlling the regret. Algorithms used both in SCO and OCO follow a vast literature on stochastic approximation methods, e.g. [28, 21, 24]. For this work, we are particularly interested in the more recent literature on acceleration in SCO [8, 15, 17]. In this research field efficiency is traditionally measured in terms of excess risk. On the one hand, regret upper bounds can be converted into excess risk bounds, through the so-called *online-to-batch conversions* [3]; on the other hand, excess risk guarantees do not directly lead to regret bounds, and even if they do some key features of the rates might be lost. These latter methods, known as *batch-to-online conversions* are discussed in Appendix D.

**Outline** In Section 2, after setting up notation and basic definitions, we introduce the *stochastically extended adversarial* model, a generalization of the standard adversarial model similar to the model used in *smoothed analysis*. Our main results can be found in Section 3. In Section 4 we illustrate our results by highlighting several special cases, such as the random order model and the adversarially corrupted stochastic model. Finally, in Section 5 we set our findings into a broader context and give perspective for future work.

## 2 Setting

We recall the *online convex optimization (OCO)* problem. Here, we consider a sequence of convex functions $f_1, \ldots f_T$ defined over a closed and bounded convex set $\mathcal{X} \subseteq \mathbb{R}^d$, which become available to the learner sequentially. In the standard *adversarial model*, the learner chooses $x_t \in \mathcal{X}$ in round $t$, then function $f_t$ is revealed and the learner suffers *loss* $f_t(x_t)$. The success of the learner is measured against all fixed $u \in \mathcal{X}$. Hence, the goal of the learner is to minimize the *regret*, that is, the difference between their cumulative loss $\sum_{t=1}^T f_t(x_t)$ and that of the best fixed choice in hindsight, namely $\min_{u \in \mathcal{X}} \sum_{t=1}^T f_t(u)$.

Throughout the paper we use the notation $[T] = \{1, \ldots, T\}$. We follow the convention that $\delta_c$ denotes a Dirac measure at a point $c$, and $\| \cdot \|$ denotes the Euclidean norm.

### 2.1 Stochastically Extended Adversarial Model

We extend the aforementioned adversarial model by letting nature choose a distribution $\mathcal{D}_t$ from a set of distributions. Then the learner suffers loss $f(x_t, \xi_t)$ where $\xi_t \sim \mathcal{D}_t$. Note that if the set of distributions is sufficiently rich, this model contains the standard adversarial model and the stochastic model as special cases (see Examples 1,2).

We introduce some notation to make this more precise. Let $\mathcal{X} \subset \mathbb{R}^d$ be a closed convex set and $\Xi$ a measurable space. Define $f : \mathcal{X} \times \Xi \to \mathbb{R}$ and assume $f(\cdot, \xi)$ is convex over $\xi \in \Xi$. Suppose $\mathfrak{D}$ is a set of probability distributions over $\Xi$. For any $\mathcal{D} \in \mathfrak{D}$, we denote the gradient mean by $\nabla F^{\mathcal{D}}(x) := \mathbb{E}_{\xi \sim \mathcal{D}} [\nabla f(x, \xi)]$ and the function mean by $F^{\mathcal{D}}(x) := \mathbb{E}_{\xi \sim \mathcal{D}} [f(x, \xi)]$. Furthermore, denote by $\sigma_{\mathcal{D}}^2$ an upper bound on the variance of the gradients

$$\sigma_{\mathcal{D}}^2 = \max_{x \in \mathcal{X}} \mathbb{E}_{\xi \sim \mathcal{D}} \left[ \left\| \nabla f(x, \xi) - \nabla F^{\mathcal{D}}(x) \right\|^2 \right].$$

We introduce some shorthand notation when distributions are indexed by rounds. Given $t \in [T]$, we write $F^t$ and $\sigma_t^2$ instead of $F^{\mathcal{D}_t}(x)$ and $\sigma_{\mathcal{D}_t}^2$, respectively. Let us now introduce the *stochastically extended adversary protocol*.

**Definition 1** (Stochastically Extended Adversary (SEA))**.** *In each round $t$, the learner chooses $x_t \in \mathcal{X}$, the SEA picks $\mathcal{D}_t \in \mathfrak{D}$. The learner and the SEA then both observe a sample $\xi_t \sim \mathcal{D}_t$, and the learner suffers loss $f(x_t, \xi_t)$.*

We note that the SEA model is closely related to the adversarial model considered in the context of *smoothed analysis* [27, 10, 32]. However, in contrast to this line of work, we do not focus on

distributions with sufficient anti-concentration (c.f., Def. 1.1 in [10]). Indeed, this restriction would exclude, among others, the fully adversarial case as described below. Note also that we assume that the SEA is an adaptive adversary, which has access to the realizations $\xi_t$, so the distribution $\mathcal{D}_{t+1}$ may depend on $\xi_1, \ldots, \xi_t$. This assumption is not relevant for the fully adversarial or the i.i.d. settings. In the former, because there is no randomness, and in the latter, because there is no change in distribution. However, it is important for some of the intermediate cases, and in particular for the random order model. The SEA model contains several common settings from the literature as special cases. To illustrate this, we list some examples.

1.  **Adversarial Model:** The SEA chooses a Dirac measure $\delta_{c_t} \in \mathfrak{D}$ in each round. Then for any $\xi_t \sim \delta_{c_t}$, the SEA selects $f(\,\cdot\,, \xi_t)$, and the model reduces to an adversary selecting directly the functions $f_t(\,\cdot\,)$.
2.  **Stochastic I.I.D. Model:** The SEA chooses a fixed $\mathcal{D} \in \mathfrak{D}$ and selects $\mathcal{D}_t = \mathcal{D}$ at each round $t$.
3.  **Adversarially Corrupted I.I.D. Model:** The adversary selects an i.i.d. source $\mathcal{D}$ and perturbs the data with adversarial corruptions. This fits in our framework as follows: given a corruption level $C \geqslant 0$, the SEA chooses distributions $\mathcal{D}_t = \mathcal{D} \otimes \delta_{c_t}$ such that $\sum_{t=1}^{T} \sup_{x \in \mathcal{X}} \| \mathbb{E}_{\xi \sim \mathcal{D}}[\nabla f(x, \xi)] - \mathbb{E}_{\xi' \sim \mathcal{D}_t}[\nabla f(x, \xi')] \| \leqslant C$.
4.  **Random Order Model (ROM):** Among a fixed family of losses $\mathcal{F} = (f_i, i \in [n])$, the SEA randomly picks functions in $\mathcal{F}$ via sampling without replacement, possibly performing multiple passes over the losses and reshuffling between the passes. Formally, define $\Xi = [n]$, and $f(x, \xi) = f_\xi$ for $\xi \in \Xi$. If $t \in [nk, n(k+1)]$ is in epoch $k \in \mathbb{N}$, then the SEA chooses the distribution $\mathcal{D}_t = \mathrm{Unif}(\Xi \setminus \{\xi_s : s \in [nk+1, n(t-1)]\})$, which is uniform on the functions that have not yet been sampled in epoch $k$.

To quantify the hardness of the loss sequence, we introduce the *cumulative stochastic variance* and *adversarial variation*; we also define an average of these quantities.

We denote by $\mathbb{E}$ the expectation taken with respect to the joint distribution of $(x_1, \xi_1, \ldots, x_T, \xi_T)$. Note that the choice of the adversary $\mathcal{D}_t$ can be random itself, as it depends on the past observations (of both the player's actions and the realizations of the $\xi_t$'s). In this case, $\sigma_t$ is also a random quantity.

**Definition 2** (Cumulative Stochastic Variance and Cumulative Adversarial Variation). *Suppose the SEA chooses distributions $\mathcal{D}_1, \ldots, \mathcal{D}_T$. Recall that $\sigma_t^2$ is a shorthand for $\sigma_{\mathcal{D}_t}^2$. The cumulative stochastic variance and the cumulative adversarial variance are defined as*

$$\sigma_{[1:T]}^{(2)} = \mathbb{E}\left[ \sum_{t=1}^{T} \sigma_t^2 \right] \qquad and \qquad \Sigma_{[1:T]}^{(2)} = \mathbb{E}\left[ \sum_{t=1}^{T} \sup_{x \in \mathcal{X}} \left\| \nabla F^t(x) - \nabla F^{t-1}(x) \right\|^2 \right].$$

*We also let $\bar{\sigma}_T$ and $\bar{\Sigma}_T$ denote the square root of the average stochastic variance or adversarial variation, respectively; that is, $\bar{\sigma}_T^2 = \sigma_{[1:T]}^{(2)}/T$ and $\bar{\Sigma}_T^2 = \Sigma_{[1:T]}^{(2)}/T$.*

Note that in the special case when all $f_t$ are fully adversarial, $\bar{\sigma}_T = 0$. On the contrary, in the stochastic case, i.e., if all for each round $t$, the distribution $\mathcal{D}_t$ is equal to a fixed (but arbitrarily chosen) $\mathcal{D}$, then $\bar{\Sigma}_T = 0$. In this case, $\bar{\sigma}_T$ reduces to the common definition of the gradient variance upper bound in the SCO literature [7, 8]. If however, the SEA chooses one distribution $\mathcal{D}_i$ for the first rounds and then switches to a different distribution $\mathcal{D}_j$, then $\bar{\sigma}_T$ can only be upper bounded by $\max(\sigma_i, \sigma_j)$. This upper bound can be pessimistic, however, for some results it gives a better intuition. For this purpose we also define the *maximal stochastic variance* and *maximal adversarial variation*.

**Definition 3** (Maximal Stochastic Variance and Maximal Adversarial Variation). *Let $\sigma_{\max}^2$ be an upper bound on all variances $\sigma_t^2$. That is,*

$$\sigma_{\max}^2 = \max_{t \in [T]} \mathbb{E}\left[ \sigma_t^2 \right] \quad and \quad \Sigma_{\max}^2 = \max_{t \in [T]} \mathbb{E}\left[ \sup_{x \in \mathcal{X}} \left\| \nabla F^t(x) - \nabla F^{t-1}(x) \right\|^2 \right].$$

**Remark 4.** *As we mentioned in the introduction, the cumulative stochastic variance and the adversarial variation have some similarities with parameters in gradual variation regret bounds. For linear functions $\langle \mu_t, \cdot \rangle$, the bounds in [12] involve the parameter $\mathrm{Var}_T = \sum_{t=1}^{T} \| \mu_t - \bar{\mu}_T \|^2$ where $\bar{\mu}_T$ is*

*the average of the gradients. For OCO with general convex functions, [4] provide upper bounds on the regret in terms of the $L_p$-deviation $D_p = \sum_{t=1}^{T} \sup_{x \in \mathcal{X}} \|\nabla f_t(x) - \nabla f_{t-1}(x)\|_p^2$. In Lemmas 13 and 14 in Appendix A, we show that in the SEA framework, both of these types of bounds are generally worse than ours, and that the difference can be arbitrarily large.*

*In [26] the regret is bounded in terms of $\sum_{t=1}^{T} \|g_t - M_t\|^2$. As mentioned in the introduction, unless the loss functions are linear or the learner has knowledge of the gradient mean, $\sum_{t=1}^{T} \|g_t - M_t\|^2$ cannot directly be reduced to $\sigma_{[1:T]}^{(2)}$ or $\Sigma_{[1:T]}^{(2)}$.*

## 2.2 Assumptions

In our analysis, we will frequently use several of the following additional assumptions. Some of these were already mentioned in the introduction. We keep them all together here, for the convenience of the reader and for clear reference. For any $\mathcal{D} \in \mathfrak{D}$:

- **(A0)** the adversary has access to independent samples $\xi \sim \mathcal{D}$.
- **(A1)** the function $f(\,\cdot\,, \xi)$ is convex, and gradients are bounded by $G$ a.s. when $\xi \sim \mathcal{D}$.
- **(A2)** the expected function $F^{\mathcal{D}}$ is $L$-smooth, i.e, $\nabla F^{\mathcal{D}}$ is $L$-Lipschitz continuous.
- **(A3)** for any $x \in \mathcal{X}$, the variance $\mathbb{E}_{\xi \sim \mathcal{D}}[\|\nabla f(x, \xi) - \nabla F^{\mathcal{D}}(x)\|^2]$ is finite.
- **(A4)** the expected function $F^{\mathcal{D}}(\,\cdot\,)$ is $\mu$-strongly convex.

We assume that **(A0)** always holds. Assumptions **(A1)**, **(A2)** and **(A3)** are standard in stochastic optimization, and are similar to common assumptions for online convex optimization. There, it is typically assumed that the adversarial samples $f_t(\,\cdot\,)$ are convex (or even linear) and the gradient norms $\|\nabla f_t(\cdot)\|$ are bounded. Note that we only require gradient Lipschitz continuity and strong convexity to hold for the expected loss.

# 3 Algorithms and Regret Bounds

## 3.1 Convex Smooth Functions

We use *Optimistic Follow-the-Regularised-Leader* (OFTRL) (see, e.g., [16, 25]) to minimize regret. Let $(\eta_t)_{t \in [T]}$ be a non-decreasing and positive sequence of stepsizes, possibly tuned adaptively with the observations. At each step $t$, the learner makes an optimistic prediction $M_t \in \mathbb{R}^d$ and updates its iterates as

$$x_t = \underset{x \in \mathcal{X}}{\operatorname{argmin}} \left\{ \left\langle x, M_t + \sum_{s=1}^{t-1} g_s \right\rangle + \frac{\|x\|^2}{\eta_t} \right\}, \tag{2}$$

where we denoted by $g_t = \nabla f(x_t, \xi_t)$ the observed gradient at time $t$. To state our results, we denote by $\mathbb{E}[\,\cdot\,]$ the expectation with respect to the joint distribution of $(x_1, \xi_1, \ldots, x_T, \xi_T)$. Our objective is to bound the average regret:

$$\mathbb{E}[R_T(u)] := \mathbb{E}\left[ \sum_{t=1}^{T} \langle g_t, x_t - u \rangle \right].$$

The following theorem, proved in Appendix B.1, is our main result for convex functions.

**Theorem 5.** *Fix a user-specified parameter $\nu > 0$. Under assumptions (A1), (A2), (A3), OFTRL, with $M_t = g_{t-1}$ and adaptive step-size $\eta_t = D^2/(\nu + \sum_{s=1}^{t-1} \eta_s \|g_s - M_s\|^2)$, has regret*

$$\mathbb{E}[R_T(u)] \leqslant D\big(6\,\bar{\sigma}_T + 3\sqrt{2}\bar{\Sigma}_T\big)\sqrt{T} + \frac{3\sqrt{2}DG}{2} + \nu + \frac{1}{\nu}\big(4D^2G^2 + 9L^2D^4\big). \tag{3}$$

*The algorithm needs to know only $D$. If $G$ and $L$ are also known, one can tune $\nu = LD^2 + D^2G^2$ to get*

$$\mathbb{E}[R_T(u)] \leqslant O\big(D\big(\bar{\sigma}_T + \bar{\Sigma}_T\big)\sqrt{T} + DG + LD^2\big).$$

*Moreover, if only convexity of the individual losses holds (A1), then tuning $\nu = 2DG$ ensures the (deterministic) bound $R_T(u) \leqslant 3\sqrt{2}DG\sqrt{T} + 4DG$.*

Without prior knowledge of the smoothness parameter, the best the player can do is to tune $\nu$ according to a guessed value $L_0$. This affects the constants in the bound by an additive term of order $(L_0 + L^2/L_0)D^2$; it would be interesting to determine if this is an inevitable price to pay for the lack of knowledge of $L$. A similar discussion can be held for $G$. Note that the worst-case regret bound of order $DG\sqrt{T}$ always holds every time OFTRL is used in this article, even without smoothness. To avoid distraction, we will not recall this fact in the applications.

The algorithm and analysis dwell on two ideas: the adaptive tuning of the learning rate à la Ada-Hedge/AdaFTRL [19, 23] with optimism, together with the fact that we keep a negative Bregman divergence term in the analysis, which is crucial to obtain our bound.

**Lower Bound.** The upper bound in Theorem 5 is tight up to additive constants, as the following result shows.

**Theorem 6.** *For any learning algorithm, and for any pair of positive numbers $(\sigma, \Sigma)$ there exists a function $f : \mathcal{X} \times \Xi \to \mathbb{R}$ and a sequence of distributions satisfying assumptions (A1), (A2), (A3) with $\bar{\sigma}_T \geqslant \sigma$ and $\bar{\Sigma}_T \geqslant \Sigma$ such that*

$$\mathbb{E}\left[R_T(u)\right] \geqslant \Omega\left(D\left(\bar{\sigma}_T + \bar{\Sigma}_T\right)\sqrt{T}\right).$$

The proof, in Appendix B.2 relies on a lower bound from stochastic optimization [1, 21] together with the fact that we can construct a sequence of convex and $L$-smooth loss functions such that $\bar{\Sigma}_T$ is in the order of the gradient norms $G$. Combining these insights with the lower bound $\Omega(DG\sqrt{T})$ [23] gives the desired result.

## 3.2 Strongly Convex and Smooth Functions

Up to this point, we have only considered functions which satisfy the weaker set of assumptions (A1), (A2), (A3). In this section, we show what improvements can be achieved if strong convexity also holds, that is, if (A4) is satisfied with some known parameter $\mu > 0$. For $g_t = \nabla f(x_t, \xi_t)$, define the surrogate loss function

$$\ell_t(x) = \langle g_t, x - x_t \rangle + \frac{\mu}{2}\left\| x - x_t \right\|^2. \tag{4}$$

We use Optimistic Follow-the-Leader (OFTL) on the surrogate losses. For each step $t$, the learner makes an optimistic prediction of the next gradient $M_t \in \mathbb{R}^d$ and selects

$$x_t = \operatorname*{argmin}_{x \in \mathcal{X}}\left\{ \sum_{s=1}^{t-1} \ell_s(x) + \langle M_t, x \rangle \right\}. \tag{5}$$

The next theorem is analogous to Theorem 5 for curved losses, and will be our main tool in establishing results for strongly convex losses; see Appendix B.3 for a proof.

**Theorem 7.** *Under assumptions (A1)–(A4), the expected regret of OFTL with $M_t = \nabla f(x_{t-1}, \xi_{t-1})$ on surrogate loss functions $\ell_t$ defined in (4) is bounded as*

$$\mathbb{E}\left[R_T(u)\right] \leqslant \frac{1}{\mu}\sum_{t=1}^{T}\frac{1}{t}\left(8\sigma_{\max}^2 + 4\mathbb{E}\left[\sup_{x \in \mathcal{X}}\left\|\nabla F^t(x) - \nabla F^{t-1}(x)\right\|^2\right]\right) + \frac{4D^2L^2}{\mu}\log\left(1 + \frac{16L}{\mu}\right)$$

$$\leqslant \frac{1}{\mu}\left(8\sigma_{\max}^2 + 4\Sigma_{\max}^2\right)\log T + \frac{4D^2L^2}{\mu}\log\left(1 + \frac{16L}{\mu}\right).$$

Note that OFTL requires no tuning besides the strong convexity parameter used in the surrogate losses. In particular, it is adaptive to the smoothness $L$.

**Lower Bound** The bound in Theorem 7 is tight, as the next result, proved in Appendix B.4 shows.

**Theorem 8.** *For any learning algorithm, and for any pair of positive numbers $(\sigma, \Sigma)$ there exists a function $f : \mathcal{X} \times \Xi \to \mathbb{R}$ and a sequence of distributions satisfying assumptions (A1), (A2), (A3) and (A4) with $\sigma_{\max} \geqslant \sigma$ and $\Sigma_{\max} \geqslant \Sigma$ such that*

$$\mathbb{E}\left[R_T(u)\right] \geqslant \Omega\left(\frac{1}{\mu}\left(\sigma_{\max}^2 + \Sigma_{\max}^2\right)\log T\right).$$

# 4  Implications

We derive consequences of our results from Section 3. Further examples can be found in Appendix E.

## 4.1  Interpolating Known Results: Fully Adversarial and i.i.d. Data

A first implication of our analysis is that we recover both the adversarial and i.i.d. rates, *via a single adaptive algorithm.*

**Convex Case**  For adversarial data, $\sigma_t = 0$ for all $t$, and $\Sigma^2_{[1:T]} \leqslant G\sqrt{2T}$. Thus, Theorem 5 guarantees a bound $R_T(u) \leqslant O(DG\sqrt{T})$, which is known to be the optimal rate up to the additive constants, cf. [37] (note that the expectation does not act on the regret in this case). Simultaneously, if the data is i.i.d., then Theorem 5 guarantees that

$$\mathbb{E}\left[R_T(u)\right] \leqslant O\big(D\sigma\sqrt{T} + LD^2 + DG\big).\tag{6}$$

From standard online-to-batch conversion, this implies an excess risk for the related SCO problem of order $O(D\sigma/\sqrt{T} + D(LD + G)/T)$, which matches the well-known result by [7] up to lower order terms. On the other hand, using batch-to-online conversion (see Appendix D) with the best known accelerated convergence result in SCO, gives $O(D\sigma\sqrt{T} + LD^2)$ regret. Therefore, up to a constant, our result coincides with the best-known results from SCO. Note that also generalizes the improvement obtained for linear functions in the i.i.d. setting in [25, Section 6.2].

**Strongly Convex Case**  The adaptive interpolation between i.i.d. and adversarial rates also holds in the strongly convex case. For adversarial data, the bound of Theorem 7 is of order $(G^2/\mu) \log T$, which is known to be the optimal worst-case rate, cf. [13]. For i.i.d. data, the dependence on $G^2$ improves to $\sigma^2$, yielding a bound of order $O((\sigma^2/\mu) \log T + LD^2\kappa \log \kappa)$. This improvement is akin to improvements obtained by *accelerated stochastic gradient descent* in the context of stochastic optimization [8, 17]. In fact, applying batch-to-online conversions and summing the optimization rates would yield a regret bound similar to ours; c.f. Appendix D.

## 4.2  Adversarially Corrupted Stochastic Data

We consider a natural generalization to online convex optimization of the corruption model considered in the bandit literature [29, 36], also recently studied in [14] for prediction with expert advice. There, the author obtains a regret bound that is the sum of the i.i.d. rate and of a term of order $\sqrt{C}$ where $C$ is the total amount of perturbation. They then raise the open question of whether similar results could be obtained for general convex losses. We provide a positive answer to this question in this section, with the regret bound in Corollary 9.

In this model, the generating process of the losses is decomposed as a combination of losses coming from i.i.d. data, with a small additive adversarial perturbation. This fits in the framework by setting $\xi_t = (\xi_{\text{iid},t}, c_t) \sim \mathcal{D}_t = \mathcal{D} \otimes \delta_{c_t}$ and

$$f(x, \xi_t) = h(x, \xi_{\text{iid},t}) + c_t(x)$$

where $c_t$ is the adversarial part of the losses selected by the adversary, and $\xi_{\text{iid},t} \sim \mathcal{D}$ is a sequence of identically distributed random variables. Note that, similarly to our inspirations [14, 29], and contrary to other corruption models for prediction with expert advice [2], we measure the regret against the perturbed data. Define $F = \mathbb{E}_{\xi \sim \mathcal{D}}[h(\cdot, \xi)]$, so that $F^t(x) = F(x) + c_t(x)$. The amount of perturbation is measured by a parameter $C > 0$ bounding

$$\sum_{t=1}^{T} \max_{x \in \mathcal{X}} \|\nabla c_t(x)\| \leqslant C \,,$$

which is a natural measure of perturbation on the feedback used by the player (note that adding a constant to the perturbations does not change the regret). In this case, the adversarial perturbation on the loss does not affect the variance and $\sigma^2_{\mathcal{D}_t} = \sigma^2_{\mathcal{D}}$. The perturbation appears in the loss variation as for any $t \geqslant 2$, for any $x \in \mathcal{X}$,

$$\|\nabla F^t(x) - \nabla F^{t-1}(x)\|^2 \leqslant 2G\|\nabla F^t(x) - \nabla F^{t-1}(x)\|$$
$$= 2G\|\nabla c_t(x) - \nabla c_{t-1}(x)\| \leqslant 2G\big(\|\nabla c_t(x)\| + \|\nabla c_{t-1}(x)\|\big) \,.$$

Upon taking the supremum over $x \in \mathcal{X}$ and summing over $t$, we get (with the convention that $c_0 \equiv 0$),

$$\Sigma_{[1:T]}^{(2)} = \sum_{t=1}^{T} \sup_{x \in \mathcal{X}} \|\nabla c_t(x) - \nabla c_{t-1}(x)\|^2 \leqslant 4GC.$$

Hence, Theorem 5 combined with the bounds on $\bar{\sigma}_T$ and $\bar{\Sigma}_T$ yields the following regret guarantee.

**Corollary 9.** *In the adversarially corrupted stochastic model, adaptive OFTRL enjoys the bound*

$$\mathbb{E}\left[R_T(u)\right] = O\left(D\sigma\sqrt{T} + D\sqrt{GC}\right).$$

This regret bound is the sum of the i.i.d. rate for the unperturbed source with a term sublinear in the amount of perturbations $C$, achieved without prior knowledge of $C$. This provides an answer to a question by [14]. The construction of the lower bound Theorem 6 also provides a partial lower bound for the corruption model, proving some worst-case optimality of our upper bound, see Appendix B.2.1.

An interesting open question that remains would be to extend these results to strongly convex losses.

**Regret against unperturbed losses** Depending on modeling choices, one might consider the perturbations $c_t$ to be a true part of the data that deviate from an i.i.d. model, or one may see them as adversarial noise which should be discarded. In the latter case, regret against the i.i.d. part of the data is most relevant. As noticed by [2, Observation 4] and [14, Remark 1], regret against the unperturbed losses and against the perturbed losses are closely related. Indeed, Corollary 9 implies the bound

$$\mathbb{E}\left[\sum_{t=1}^{T} F(x_t) - F(u)\right] \leqslant \mathbb{E}[R_T(u)] + 2D\mathbb{E}\left[\sum_{t=1}^{T} \|\nabla c_t(w_t)\|\right] \leqslant \mathcal{O}(D\sigma\sqrt{T} + D\sqrt{GC} + 2DC).$$

## 4.3 Random Order Models

We apply our results from Section 3 to the Random Order model. The online ROM was introduced by [6] as a way of restricting the power of the adversary in OCO. Note that the ROM is a popular model in combinatorial online optimization, e.g., in the context of profit maximization [9]. Our results highlight that the rates in the ROM model, which is not i.i.d., are almost the same as the rates of the i.i.d. model obtained via sampling in the same set of losses with replacement.

**Corollary 10.** *In the single-pass ROM with convex and $L$-smooth losses $(f_k)_{k \in [T]}$, OFTRL (c.f. (2)) enjoys the regret bound*

$$\mathbb{E}[R_T(u)] \leqslant O\left(D\sigma_1\sqrt{\log\left(e\frac{\widetilde{\sigma}_1}{\sigma_1}\right)T} + LD^2 + DG\right),$$

*where*

$$\sigma_1^2 = \max_{x \in \mathcal{X}} \frac{1}{T}\sum_{t=1}^{T}\left\|\nabla f_t(x) - \frac{1}{T}\sum_{s=1}^{T}\nabla f_s(x)\right\|^2 \text{ and } \widetilde{\sigma}_1^2 = \frac{1}{T}\sum_{t=1}^{T}\max_{x \in \mathcal{X}}\left\|\nabla f_t(x) - \frac{1}{T}\sum_{s=1}^{T}\nabla f_s(x)\right\|^2.$$

We remark that $\sigma_1 \leqslant \widetilde{\sigma}_1 \leqslant \min\{4G^2, T\sigma_1\}$. Hence, the logarithm of the ratio $\log(\widetilde{\sigma}_1/\sigma_1)$ is always under control. The proof of Corollary 10 consists in controlling the adversarial variation and the cumulative variance thanks to the following lemma, proved in Appendix C.1.

**Lemma 11.** *In the single-pass ROM, we have $\Sigma_{[1:T]}^{(2)} \leqslant 8G^2$ and $\bar{\sigma}_{[1:T]}^{(2)} \leqslant T\sigma_1^2 \log(2e^2\widetilde{\sigma}_1^2/\sigma_1^2)$.*

We would like to emphasize that our results are complementary to those of [6, 31]. The focus of these works is to relax the assumption that individual losses are convex, and to only require convexity of the average loss function, leading to very different technical challenges. Inquiring if our results can also be achieved under the weaker assumptions of [31] would be an interesting direction for future work.

We also consider the multi-pass ROM. Let $P \in \mathbb{N}$ denote the number of passes. From Lemma 11 and Corollary 10 we directly obtain

$$\mathbb{E}\left[R_T(u)\right] \leqslant \mathcal{O}\left(D\sigma_1\sqrt{\log\left(e\frac{\widetilde{\sigma}_1}{\sigma_1}\right)T} + DG\sqrt{P} + LD^2\right).$$

Combining Lemma 11 with Theorem 7 also gives the following corollary for strongly convex functions; see Appendix C.2 for a proof.

**Corollary 12.** *Under the same assumption as in Theorem 7, the expected regret of the ROM is bounded by*

$$\mathbb{E}\left[R_T(u)\right] \leqslant O\left(\frac{\sigma_1^2}{\mu}\log T + \frac{G^2}{\mu} + LD^2\kappa\log\kappa\right).$$

*For multi-pass ROM with $P$ passes, we obtain*

$$\mathbb{E}\left[R_T(u)\right] \leqslant O\left(\frac{\sigma_1^2}{\mu}\log T + \frac{G^2}{\mu} + \frac{G^2\log P}{n\mu} + LD^2\kappa\log\kappa\right).$$

## 5   Conclusion and future work

As we showed, the exploitation of smoothness of the expected loss functions reduces the dependence of the regret bound on the maximal gradient norm to a dependence on the cumulative stochastic variance and the adversarial variation. Furthermore, we took a step towards a deeper theoretical understanding of the practically relevant intermediate scenarios. Our approach also opens several interesting new research directions. For instance, in the ROM, as mentioned in Section 4.3, an interesting question is whether a regret bound with dependence on $\sigma$ instead of $G$ can also be achieved with weaker assumptions as in [31].

Another interesting question is whether it is possible to unify the analyses and algorithms for the strongly convex and convex case. So far our analyses of these cases were intrinsically different and the choice of the algorithm requires the knowledge of strong convexity constant $\mu$. Since this knowledge might not be available, it is of practical interest to design an adaptive method that can automatically get the best rate without manually tuning for $\mu$. Furthermore, it would be interesting to investigate if similar or better improvements of the regret bound can be achieved with different optimistic predictions $M_t$. Our matching lower bounds show that there are instances where $M_t = g_{t-1}$ captures the right adversarial behavior. However, there are cases where other measures of regularity would be more relevant. One such example would be an adversary oscillating between two distributions. Interestingly, using $M_t = g_{t-2}$ would then recover optimal bounds. An interesting future research direction would be to investigate what can be achieved when using linear combinations of past gradients as optimistic predictions.

Social impact: this work is theoretical and does not give rise to any direct societal concerns.

## Acknowledgments and Disclosure of Funding

Sachs, Hadiji, and Van Erven were supported by the Netherlands Organization for Scientific Research (NWO) under grant number VI.Vidi.192.095. Guzmán's research is partially supported by the INRIA Associate Teams project, FONDECYT 1210362 grant and ANID Anillo ACT210005 grant, and National Center for Artificial Intelligence CENIA FB210017, Basal ANID. Part of this work was done while CG was at the University of Twente.

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
