# A  Proofs of Section 2

Note that $\mathrm{Var}_T = \sum_{t=1}^T \|\nabla f_t(x) - \mu_T^*\|^2$ can be understood as an empirical approximation of $\sigma_{[1:T]}^{(2)}$. The following lemma shows the relation of $\mathrm{Var}_T$ to parameters $\sigma_{[1:T]}^{(2)}$ and $\Sigma_{[1:T]}^{(2)}$.

**Lemma 13.** *Define* $\mathrm{Var}_T = \sum_{t=1}^T \|\nabla f_t(x) - \mu_T^*\|^2$ *with* $\mu_T^* = \frac{1}{T} \sum_{t=1}^T \nabla f_t(x)$. *In expectation with respect to distributions* $\mathcal{D}_1, \dots \mathcal{D}_T$,

$$\mathbb{E}\left[\mathrm{Var}_T\right] \geqslant \frac{1}{5} \left( \sigma_{[1:T]}^{(2)} + \Sigma_{[1:T]}^{(2)} \right).$$

*Furthermore, there exists distributions such that* $\mathbb{E}\left[\mathrm{Var}_T\right]$ *is arbitrarily larger than* $\sigma_{[1:T]}^{(2)} + \Sigma_{[1:T]}^{(2)}$.

*Proof.* Since the distribution mean minimizes the least squares error, $\mathbb{E}\left[\mathrm{Var}_T\right] \geqslant \sigma_{[1:T]}^{(2)}$ always holds. Using the same argument, we have

$$\Sigma_{[1:T]}^{(2)} = \sum_{t=1}^T \left\|\nabla F^t - \nabla F^{t-1}\right\|^2 \leqslant 4 \sum_{t=1}^T \left\|\nabla f_t - \nabla F^t\right\|^2 \leqslant 4 \sum_{t=1}^T \left\|\nabla f_t - \mu_T^*\right\|^2 = 4\,\mathrm{Var}_T \,.$$

Consider $\mathcal{X} = [-1, 1]$ and $f(x, \xi) = \xi x$. Now suppose the SEA chooses truncated normal distribution with mean $-2$ for the first $T/2$ rounds, then truncated normal distribution with mean $2$ for the remaining rounds. Assume in both cases the variance is $\sigma^2$ and truncation is in the range $[-G, G]$ for $G > 0$. Hence, for sufficiently large $T$, $\mu_T = 0$ and $\mathrm{Var}_T = \sum_{t=1}^T \|f_t\|^2 \propto T G^2$. However, $\sigma_{[1:T]}^{(2)}$ is equal to $T\sigma^2$ which can be considerably smaller than $TG^2$. The price for the distribution switch is captured with a small constant overhead by $\Sigma_{[1:T]}^{(2)} \leqslant 2G^2$. Thus, $\sigma_{[1:T]}^{(2)} + \Sigma_{[1:T]}^{(2)} \propto T\sigma^2 + 2G^2$. $\quad\square$

**Lemma 14.** *Define* $D_p = \sum_{t=1}^T \sup_{x \in \mathcal{X}} \|\nabla f_t(x) - \nabla f_{t-1}(x)\|_p^2$. *In the SEA framework,*

$$\mathbb{E}\left[D_2\right] \geqslant \frac{1}{2} \left( \sigma_{[1:T]}^{(2)} + \Sigma_{[1:T]}^{(2)} \right).$$

*Furthermore, there exist instances such that* $\mathbb{E}\left[D_2\right] \gg \sigma_{[1:T]}^{(2)} + \Sigma_{[1:T]}^{(2)}$.

*Proof.* We shall in fact prove that $\mathbb{E}[D_2] \geqslant \max(\sigma_{[1:T]}^{(2)}, \Sigma_{[1:T]}^{(2)})$, which directly implies the first part of the statement. Fix a time $t \geqslant 2$ and let $\mathcal{G}_{t-1}$ denote the $\sigma$-algebra generated by $(x_1, \xi_1, \dots, \xi_{t-1}, x_t)$. Then for any $x \in \mathcal{X}$, the variable $F^t(x) = \mathbb{E}[f_t(x)|\mathcal{G}_{t-1}]$ is $\mathcal{G}_{t-1}$-measurable, and we have

$$\mathbb{E}\left[\sup_{x \in \mathcal{X}} \|\nabla f_t(x) - \nabla f_{t-1}(x)\|^2 \,\Big|\, \mathcal{G}_{t-1}\right] \geqslant \sup_{x \in \mathcal{X}} \mathbb{E}\left[\|\nabla f_t(x) - \nabla f_{t-1}(x)\|^2 \mid \mathcal{G}_{t-1}\right]$$

$$\geqslant \sup_{x \in \mathcal{X}} \mathbb{E}\left[\|\nabla f_t(x) - \nabla F^t(x)\|^2 \mid \mathcal{G}_{t-1}\right] = \sigma_t^2$$

since $f_{t-1} = f(\cdot, \xi_{t-1})$ is $\mathcal{G}_{t-1}$-measurable, and $\nabla F^t(x) = \mathbb{E}[\nabla f^t(x) \mid \mathcal{G}_{t-1}]$. Therefore, by conditioning on $\mathcal{G}_{t-1}$ at time step $t$, and applying the tower rule, we obtain

$$\mathbb{E}\left[\sum_{t=1}^T \sup_{x \in \mathcal{X}} \|\nabla f_t(x) - \nabla f_{t-1}(x)\|^2\right] \geqslant \mathbb{E}\left[\sum_{t=1}^T \sigma_t^2\right] = \sigma_{[1:T]}^{(2)} \,.$$

The lower bound by $\Sigma_{[1:T]}^{(2)}$ holds by a direct application of Jensen's inequality, and by swapping suprema and expectations.

For the second part of the lemma, consider the $d$-dimensional euclidean ball $\mathcal{X} = B_d(1) \subset \mathbb{R}^d$, and $\Xi = [d]$. Define

$$f(x, i) = x_i^2/2 \,.$$

Consider a fully stochastic (i.i.d.) SEA picking $\xi_t \in [d]$ uniformly at random at every time step. Then $\Sigma_{[1:T]}^{(2)} = 0$. We shall now see that $\sigma_{[1:T]}^{(2)} \leqslant T/d$. Indeed, for any $x \in \mathcal{X}$ and $t \in [T]$, then $F^t$ does not depend on $t$ and its value is

$$F(x) = \mathbb{E}_{I \sim \mathcal{D}}[f(x, I)] = \frac{1}{2d} \sum_{i=1}^{d} x_i^2 \,,$$

which is a convex and smooth function. We can upper bound the variance, as for any $x \in \mathcal{X}$,

$$\mathbb{E}_{I \sim \mathcal{D}}\big[\|\nabla f(x, I) - \nabla F(x)\|^2\big] \leqslant \mathbb{E}_{I \sim \mathcal{D}}\big[\|\nabla f(x, I)\|^2\big] = \mathbb{E}_{I \sim \mathcal{D}}\big[\|x_I e_I\|^2\big] = \frac{1}{d} \sum_{i=1}^{d} x_i^2 \leqslant \frac{1}{d}\,.$$

Therefore, after the taking the supremum over $x \in \mathcal{X}$, we see that $\sigma_t^2 \leqslant 1/d$. On the other hand, for any $I, J \in [d]$, we have

$$\|\nabla f(x, I) - \nabla f(x, J)\|^2 = \|x_I e_I - x_J e_J\|^2 = (x_I^2 + x_J^2)\mathbf{1}\{I \neq J\}\,.$$

The maximum in the ball of this difference is reached at $x = \sqrt{2}/2(e_I + e_J)$ and

$$\max_{x \in \mathcal{X}} \|\nabla f(x, I) - \nabla f(x, J)\|^2 = \mathbf{1}\{I \neq J\}\,.$$

Therefore, if $I$ and $J$ are independent and uniformly distributed over $[d]$, then

$$\mathbb{E}_{(I,J) \sim \mathcal{D} \otimes \mathcal{D}}\Big[ \max_{x \in \mathcal{X}} \|\nabla f(x, I) - \nabla f(x, J)\|^2 \Big] = \mathbb{P}_{(I,J) \sim \mathcal{D} \otimes \mathcal{D}}[I \neq J] = \left(1 - \frac{1}{d}\right)^2 \geqslant 1/4\,.$$

Summarizing the above inequalities, we have built an example in which

$$\mathbb{E}[D_2] \geqslant \frac{T}{4} \gg \frac{T}{d} \geqslant \sigma_{[1:T]}^{(2)} + \Sigma_{[1:T]}^{(2)}\,.$$

In particular, the expectation of the variation $D_2$ can be arbitrarily larger than the cumulative variance, and our bounds are then tighter than those obtained via a direct application of known results. $\qquad \square$

## B  Proofs of Section 3

### B.1  Proof of Theorem 5

To prove Theorem 5, we need the following well-known result from the literature.

**Lemma 15.** *Suppose $f_t(\,\cdot\,)$ are convex for all $t \in [T]$ and $\psi_t(\,\cdot\,) = \frac{2}{\eta_t}\|\cdot\|^2$. Further, let $g_t \in \partial f_t(x_t)$ and assume $\eta_t \geqslant \eta_{t+1}$. Then the regret for OFTRL is bounded by*

$$R_T \leqslant \frac{D^2}{\eta_T} + \sum_{t=1}^{T} \left( \langle g_t - M_t, x_t - x_{t+1} \rangle - \frac{1}{\eta_t}\|x_{t+1} - x_t\|^2 \right). \tag{7}$$

*Proof.* Denote $F_t(x) := \psi_t(x) + \sum_{s=1}^{t-1} f_s(x)$. Note that $F_t$ is $\frac{2}{\eta_t}$-strongly convex. Thus, Thm 7.29 in [22] gives

$$R_T(u) \leqslant \psi_{T+1}(u) - \psi_1(x_1)$$

$$+ \sum_{t=1}^{T} \left( \langle g_t - M_t, x_t - x_{t+1} \rangle - \frac{1}{\eta_t}\|x_t - x_{t+1}\|^2 + \psi_t(x_{t+1}) - \psi_{t+1}(x_{t+1}) \right).$$

Since $\psi_t(x) - \psi_{t+1}(x) \leqslant 0$ and $\psi_{T+1}(u) \leqslant \frac{D^2}{\eta_T}$, this gives

$$\leqslant \frac{D^2}{\eta_T} + \sum_{t=1}^{T} \left( \langle g_t - M_t, x_t - x_{t+1} \rangle - \frac{1}{\eta_t}\|x_{t+1} - x_t\|^2 \right). \qquad \square$$

**Theorem 5.** *Fix a user-specified parameter $\nu > 0$. Under assumptions (A1) ,(A2) ,(A3) , OFTRL, with $M_t = g_{t-1}$ and adaptive step-size $\eta_t = D^2/(\nu + \sum_{s=1}^{t-1} \eta_s \|g_s - M_s\|^2)$, has regret*

$$\mathbb{E}\left[R_T(u)\right] \leqslant D\left(6\,\bar{\sigma}_T + 3\sqrt{2}\bar{\Sigma}_T\right)\sqrt{T} + \frac{3\sqrt{2}DG}{2} + \nu + \frac{1}{\nu}\left(4D^2G^2 + 9L^2D^4\right). \qquad (3)$$

*The algorithm needs to know only $D$. If $G$ and $L$ are also known, one can tune $\nu = LD^2 + D^2G^2$ to get*

$$\mathbb{E}\left[R_T(u)\right] \leqslant O\left(D\left(\bar{\sigma}_T + \bar{\Sigma}_T\right)\sqrt{T} + DG + LD^2\right).$$

*Moreover, if only convexity of the individual losses holds (A1) , then tuning $\nu = 2DG$ ensures the (deterministic) bound $R_T(u) \leqslant 3\sqrt{2}DG\sqrt{T} + 4DG$ .*

*Proof of Theorem 5.* Write $g_t = \nabla f(x_t, \xi_t)$ and denote by $\mathbb{E}$ the expectation with respect to all the randomness. Using the Optimistic FTRL bound from Lemma 15,

$$\sum_{t=1}^{T}\langle g_t, x_t - u\rangle \leqslant \frac{D^2}{\eta_T} + \sum_{t=1}^{T}\left(\langle g_t - M_t,\, x_t - x_{t+1}\rangle - \frac{1}{2\eta_t}\|x_t - x_{t+1}\|^2\right)$$

$$\leqslant \frac{D^2}{\eta_T} + \sum_{t=1}^{T}\frac{\eta_t}{2}\|g_t - M_t\|^2 - \sum_{t=1}^{T}\frac{1}{2\eta_t}\|x_t - x_{t+1}\|^2,$$

where the last inequality is obtained by separating the negative norm term in two parts, and keeping half of it in the regret bound. Let us plug in the value $\eta_t$,

$$\eta_t = D^2\left(\nu + \sum_{s=1}^{t-1}\eta_s\|g_s - M_s\|^2\right)^{-1},$$

and use the fact that $\eta_t \leqslant D^2/C$ to further upper bound the deterministic regret by

$$\nu + \frac{3}{2}\sum_{t=1}^{T}\eta_t\|g_t - M_t\|^2 - \frac{\nu}{2D^2}\sum_{t=1}^{T}\|x_t - x_{t+1}\|^2. \qquad (8)$$

To bound the second term above, we first compute

$$\left(\sum_{t=1}^{T}\eta_t\|g_t - M_t\|^2\right)^2 = \sum_{t=1}^{T}\sum_{s=1}^{T}\eta_s\|g_s - M_s\|^2\eta_t\|g_t - M_t\|^2$$

$$= 2\sum_{t=1}^{T}\left(\sum_{s=1}^{t-1}\eta_s\|g_s - M_s\|^2\right)\eta_t\|g_t - M_t\|^2 + \sum_{t=1}^{T}\eta_t^2\|g_t - M_t\|^4.$$

Since $\eta_t \leqslant D^2/(\sum_{s=1}^{t-1}\eta_s\|g_s - M_s\|^2)$

$$\leqslant 2D^2\sum_{t=1}^{T}\|g_t - M_t\|^2 + \left(\frac{\sum_{t=1}^{T}\eta_t^2\|g_t - M_t\|^4}{\sum_{t=1}^{T}\eta_t\|g_t - M_t\|^2}\right)\sum_{t=1}^{T}\eta_t\|g_t - M_t\|^2.$$

Now we use the fact $X^2 \leqslant 2A + BX$ implies $X \leqslant \sqrt{2A} + B$ for $A, B > 0$. Hence

$$\sum_{t=1}^{T}\eta_t\|g_t - M_t\|^2 \leqslant D\sqrt{2\sum_{t=1}^{T}\|g_t - M_t\|^2 + \frac{\sum_{t=1}^{T}\eta_t^2\|g_t - M_t\|^4}{\sum_{t=1}^{T}\eta_t\|g_t - M_t\|^2}}.$$

Next we use $\eta_t \leqslant D^2/\nu$ and $\|g_t - M_t\|^2 \leqslant 4G^2$ to bound the last term. The sum satisfies $\sum_{t=1}^{T}\eta_t^2\|g_t - M_t\|^4 \leqslant (4G^2D^2/\nu)\sum_{t=1}^{T}\eta_t\|g_t - M_t\|^2$ and we can further bound the term above

$$\leqslant D\sqrt{2\sum_{t=1}^{T}\|g_t - M_t\|^2 + \frac{4D^2G^2}{\nu}}.$$

All in all, we have

$$\sum_{t=1}^{T} \langle g_t, x_t - u \rangle \leqslant \frac{3\sqrt{2}D}{2} \sqrt{\sum_{t=1}^{T} \|g_t - M_t\|^2 + \nu} + \frac{4D^2G^2}{\nu} - \frac{\nu}{2D^2} \sum_{t=1}^{T} \|x_{t+1} - x_t\|^2. \quad (9)$$

Note that we have not used any assumption on the expected $F^t$'s, and in particular not the smoothness. Therefore, even if the expected losses are not smooth, our analysis already entails that if $\|M_t\| \leqslant G$

$$\sum_{t=1}^{T} \langle g_t, x_t - u \rangle \leqslant \frac{3\sqrt{2}DG}{2} \sqrt{4T} + \nu + \frac{4D^2G^2}{\nu}, \quad (10)$$

proving the final claim of the statement.

Let us now proceed with the proof of the finer results. We use the value of $M_t$, together with the fact that (by convexity of $a \mapsto \|a\|^2$), for any $t \geqslant 2$,

$$\|g_t - g_{t-1}\|^2 \leqslant 4 \|g_t - \nabla F^t(x_t)\|^2 + 4 \|\nabla F^t(x_t) - \nabla F^t(x_{t-1})\|^2$$
$$+ 4 \|\nabla F^t(x_{t-1}) - \nabla F^{t-1}(x_{t-1})\|^2 + 4 \|\nabla F^{t-1}(x_{t-1}) - g_{t-1}\|^2$$
$$\leqslant 4 \|g_t - \nabla F^t(x_t)\|^2 + 4L^2 \|x_t - x_{t-1}\|^2$$
$$+ 4 \|\nabla F^t(x_{t-1}) - \nabla F^{t-1}(x_{t-1})\|^2 + 4 \|\nabla F^{t-1}(x_{t-1}) - g_{t-1}\|^2.$$

Therefore, using the inequality $\sqrt{a+b} \leqslant \sqrt{a} + \sqrt{b}$, as well as $\|g_1\| \leqslant G$ and reorganizing the terms,

$$\sqrt{\sum_{t=1}^{T} \|g_t - g_{t-1}\|^2} \leqslant G + \sqrt{8 \sum_{t=2}^{T} \|g_t - \nabla F^t(x_t)\|^2}$$
$$+ 2L \sqrt{\sum_{t=2}^{T} \|x_t - x_{t-1}\|^2} + 2\sqrt{\sum_{t=2}^{T} \|\nabla F^t(x_{t-1}) - \nabla F^{t-1}(x_{t-1})\|^2}.$$

The sum of the variations of $x_t$'s can be canceled thanks to the negative term in (9), as

$$\frac{3\sqrt{2}D}{2} 2L \sqrt{\sum_{t=1}^{T-1} \|x_{t+1} - x_t\|^2} - \frac{\nu}{2D^2} \sum_{t=1}^{T} \|x_{t+1} - x_t\|^2 \leqslant \sup_{X \geqslant 0} \left\{ 3\sqrt{2}LDX - \frac{\nu}{2D^2} X^2 \right\} = \frac{9L^2D^4}{\nu}.$$

After replacing these bounds in (9), we have obtained the regret bound

$$\sum_{t=1}^{T} \langle g_t, x_t - u \rangle \leqslant 6D \sqrt{\sum_{t=1}^{T} \|g_t - \nabla F^t(x_t)\|^2} + 3\sqrt{2}D \sqrt{\sum_{t=1}^{T-1} \|\nabla F^{t+1}(x_t) - \nabla F^t(x_t)\|^2}$$
$$+ \frac{3\sqrt{2}DG}{2} + \nu + \frac{4D^2G^2}{\nu} + \frac{9L^2D^4}{\nu}. \quad (11)$$

We will then take expectations in the inequality above. To bound the right-hand side, let us denote by $\mathcal{G}_t = \sigma(x_1, \xi_1, \ldots, x_{t-1}, \xi_{t-1}, x_t)$, then $\xi_t$ is distributed according to $\mathcal{D}_t$ given $\mathcal{G}_t$, and since $x_t$ is $\mathcal{G}_t$-measurable, therefore

$$\mathbb{E}\big[\|\nabla f(x_t, \xi_t) - \nabla F^t(x_t)\|^2 \mid \mathcal{G}_t\big] \leqslant \mathbb{E}_{\xi \sim \mathcal{D}_t}\big[\|\nabla f(x_t, \xi) - \nabla F^t(x_t)\|^2\big]$$
$$\leqslant \sup_{x \in \mathcal{X}} \mathbb{E}_{\xi \sim \mathcal{D}_t}\big[\|\nabla f(x, \xi) - \nabla F^t(x)\|^2\big] = \sigma_t^2.$$

Therefore, by the tower rule,

$$\mathbb{E}\big[\|\nabla f(x_t, \xi_t) - \nabla F^t(x_t)\|^2\big] = \mathbb{E}\Big[\mathbb{E}\big[\|\nabla f(x_t, \xi_t) - \nabla F^t(x_t)\|^2 \mid \mathcal{G}_t\big]\Big] \leqslant \mathbb{E}\big[\sigma_t^2\big]. \quad (12)$$

The final result follows from taking expectations in (11), applying Jensen's inequality, incorporating (12) and using the definitions of $\bar{\sigma}_T$ and $\bar{\Sigma}_T$. $\qquad\square$

## B.2 Proof of Theorem 6

**Theorem 6.** *For any learning algorithm, and for any pair of positive numbers $(\sigma, \Sigma)$ there exists a function $f : \mathcal{X} \times \Xi \to \mathbb{R}$ and a sequence of distributions satisfying assumptions (A1) , (A2) ,(A3) with $\bar{\sigma}_T \geqslant \sigma$ and $\bar{\Sigma}_T \geqslant \Sigma$ such that*

$$\mathbb{E}\left[R_T(u)\right] \geqslant \Omega\big(D\left(\bar{\sigma}_T + \bar{\Sigma}_T\right)\sqrt{T}\big).$$

*Proof of Theorem 6.* Suppose we are given two parameters $\hat{\sigma}_T$ and $\hat{\Sigma}_T$, we show that there exists a sequence of distributions $\mathcal{D}_1, \ldots \mathcal{D}_T$ such that the expected regret is at least $\Omega(D(\hat{\sigma}_T + \hat{\Sigma}_T)\sqrt{T})$. Let $1 \leqslant a < b$ be constants such that $a \geqslant \frac{1}{2}b$. Since for any closed convex set, there exist an affine transformation which maps it to the interval $[a, b]$, we assume without loss of generality that $\mathcal{X} = [a, b]$.

Suppose $f : \mathcal{X} \times \Xi \to \mathbb{R}$ and let $z^{(\sigma)}, z^{(\Sigma)} \in \mathbb{R}$. Assume the gradients have the form

$$\nabla f(x, \xi) = z^{(\sigma)} \qquad \text{or} \qquad \nabla f(x, \xi) = z^{(\Sigma)}.$$

Assume SEA chooses each case with probability $1/2$. The idea is to construct two sequences $\{z_t^{(\sigma)}\}_{t \in [T]}$ and $\{z_t^{(\Sigma)}\}_{t \in [T]}$ such that these sequences have at least $\Omega(D\hat{\sigma}_T\sqrt{T})$ and $\Omega(D\hat{\Sigma}_T\sqrt{T})$ expected regret, respectively. Therefore, let $x_t$ denote the learners choice in round $t$ and define linearised regret with respect to $\{z_t^{(\sigma)}\}_{t \in [T]}$ and $\{z_t^{(\Sigma)}\}_{t \in [T]}$.

$$R_T^\sigma = \min_{u \in \mathcal{X}} \sum_{t \in [T]} \left\langle z_t^{(\sigma)}, x_t - u \right\rangle \quad \text{and} \quad R_T^\Sigma = \min_{u \in \mathcal{X}} \sum_{t \in [T]} \left\langle z_t^{(\Sigma)}, x_t - u \right\rangle.$$

**Case $R_T^\Sigma$:** Let $G = \hat{\Sigma}_T$. Define $z : \mathcal{X} \to \mathbb{R}$, $z(x) = \frac{1}{4b}Gx^2$. Then $z$ is $G$-Lipschitz, smooth and $z'(x) \in [\frac{1}{2}G, G]$ for any $x \in \mathcal{X}$. Let $\{\varepsilon_t\}_{t \in [T]}$ be an i.i.d. sequence of Rademacher random variables, that is, $\mathbb{P}\left[\varepsilon_t = 1\right] = \mathbb{P}\left[\varepsilon_t = -1\right] = 1/2$. The sequence $\{z_t^{(\Sigma)}\}_{t \in [T]}$ is defined as

$$z_t^{(\Sigma)} = \begin{cases} 0 & \text{if } t \text{ even} \\ \varepsilon_t\, z'(x_t) & \text{if } t \text{ odd}. \end{cases}$$

Using that $\mathbb{E}\left[\varepsilon\right] = 0$ together with the definition of $z_t^{(\Sigma)}$ gives

$$\mathbb{E}_{\varepsilon \sim \text{Rad}}\left[R_T^\Sigma\right] = \mathbb{E}_{\varepsilon \sim \text{Rad}}\left[\min_{u \in \mathcal{X}} \sum_{t=1}^T \left\langle z_t^{(\Sigma)}, x_t - u \right\rangle\right]$$

$$= \mathbb{E}_{\varepsilon \sim \text{Rad}}\left[\max_{u \in \mathcal{X}} \sum_{\substack{t=1 \\ t \text{ odd}}}^T \left\langle \varepsilon_t\, z'(x_t), u \right\rangle\right]$$

$$\geqslant \frac{G}{4}\, \mathbb{E}_{\varepsilon \sim \text{Rad}}\left[\max_{u \in \mathcal{X}} \sum_{t=1}^{\frac{T}{2}} \varepsilon_t u\right].$$

Now use that for a linear function $l(x)$, $\max_{x \in [a,b]} l(x) = \max_{x \in \{a,b\}} l(x) = \frac{l(a+b)}{2} + \frac{|l(a-b)|}{2}$.

$$= \frac{G}{4}\, \mathbb{E}_{\varepsilon \sim \text{Rad}}\left[\max_{u \in \{a,b\}} \sum_{t=1}^{\frac{T}{2}} \varepsilon_t u\right]$$

$$= \frac{G}{8}\, \mathbb{E}_{\varepsilon \sim \text{Rad}}\left[\sum_{t=1}^{T/2} \varepsilon_t(a+b)\right] + \frac{G}{8}\, \mathbb{E}_{\varepsilon \sim \text{Rad}}\left[\left|\sum_{t=1}^{T/2} G\varepsilon_t(a-b)\right|\right]$$

$$= \frac{G}{16}\, \mathbb{E}_{\varepsilon \sim \text{Rad}}\left[\left|\sum_{t=1}^{T} \varepsilon_t(a-b)\right|\right].$$

Where we have used $\mathbb{E}[\varepsilon] = 0$ again. Now we use that by definition $D = \sup_{x,y \in \mathcal{X}} \|x - y\|$.

$$= \frac{GD}{16} \mathbb{E}_{\varepsilon \sim \text{Rad}} \left[ \left| \sum_{t=1}^{T} \varepsilon_t \right| \right] \geqslant \frac{1}{32} D \sqrt{G^2 T}.$$

In the last step, we have used the Khintchine inequality. Now note that $G^2 \geqslant \frac{1}{2} \sup_{x \in \mathcal{X}} \|z'(x)\|^2 = \frac{1}{2} \sup_{x \in \mathcal{X}} \|z'(x) - 0\|^2$. Due to the definition of the sequence $\{z_t^{(\Sigma)}\}_{t \in [T]}$, if $\|\nabla f(x, \xi^t)\| \neq 0$, then $\|\nabla f(x, \xi^{t-1})\| = 0$. Thus, $\frac{1}{2} \sup_{x \in \mathcal{X}} \|z'(x) - 0\|^2 = \frac{1}{2} \sup_{x \in \mathcal{X}} \|\nabla f(x, \xi^t) - \nabla f(x, \xi^{t-1})\|^2$ for any $t \in [T]$.

Thus, $\sqrt{G^2 T} = \sqrt{T/(2T) \sum_{t=1}^{T} \sup_{x \in \mathcal{X}} \|\nabla f(x, \xi^t) - \nabla f(x, \xi^{t-1})\|^2} = \bar{\Sigma}_T \sqrt{T/2}$. Setting the value $G = \hat{\Sigma}_T$ completes this part of the proof.

**Case $R_T^\sigma$:** We will show this part by contradiction. Suppose that $\mathcal{D}$ is a distribution such that the variance of the gradients $\sigma$ is equal to $\hat{\sigma}_T$. Suppose the SEA picks this distribution every round and assume for contradiction that $\mathbb{E}[R_T^\sigma] \leqslant o(D\sigma\sqrt{T})$. Using online-to-batch conversion gives a convergence bound of order $o(D\sigma/\sqrt{T})$ which contradicts well-known lower bounds from stochastic optimization (c.f., [1, 21] Section 5). $\square$

### B.2.1 Lower Bound for Adversarially Corrupted Stochastic Data

A careful look at the proof of Theorem 6 shows that the same construction applies to the corrupted data model, providing some worst-case optimality of our bounds.

**Proposition 16.** *For any algorithm, for any $C > 0$, and $\sigma > 0$ there exists an instance of the Adversarially Corrupted model with variance $\sigma$, corruption level $C$, and gradients of norm $G = C/T$ such that*

$$\mathbb{E}[R_T] \geqslant \Omega(D\sigma\sqrt{T} + D\sqrt{GC}).$$

*Proof.* Case I. If $\sqrt{GC} \geqslant \sigma\sqrt{T}$. The first construction from the proof of Theorem 6 guarantees the existence of a sequence of linear losses with gradient $G$ against which the regret is $\Omega(DG\sqrt{T}) = \Omega(D\sqrt{GC})$. This sequence can be seen as adversarial perturbations on identically stochastic data. Case II. If $\sigma\sqrt{T} \geqslant \sqrt{GC}$. Then the stochastic lower bound in the second part of the proof of Theorem 6 provides a lower bound on the expected regret of $\Omega(D\sigma\sqrt{T})$. $\square$

### B.3 Proof of Theorem 7

We first need a well-known result for OFTL for strongly convex loss functions.

**Lemma 17.** *Suppose $f_t(\cdot)$ are $\mu$-strongly convex for all $t \in [T]$ and $\psi_t(\cdot) = 0$. Further, let $m_t : \mathcal{X} \to \mathbb{R}$ denote*

*the optimistic prediction, and $g_t \in \partial f_t(x_t)$, $M_t \in \partial m_t(x_t)$. Then the regret for OFTRL is bounded by*

$$R_T(u) \leqslant \sum_{t=1}^{T} \left( \langle g_t - M_t, x_t - x_{t+1} \rangle - \frac{t\mu}{2} \|x_t - x_{t+1}\|^2 \right).$$

This is a well-known result and can be found in the literature, e.g., [22]. We include a short proof for completeness.

*Proof.* Let $\bar{F}_t(x) = \sum_{s=1}^{t-1} f_t(x)$ and $G_t \in \partial \bar{F}_{t+1}(x_t)$. Note that $\bar{F}_t$ is $[(t-1)\mu]$-strongly convex. From standard analysis (see, e.g., [22] Lem. 7.1) we obtain

$$\sum_{t=1}^{T}[f_t(x_t) - f_t(u)] = \underbrace{\bar{F}_{T+1}(x_{T+1}) - \bar{F}_{T+1}(u)}_{\leqslant 0} + \sum_{t=1}^{T}[\underbrace{\bar{F}_t(x_t) + f_t(x_t)}_{=\bar{F}_{t+1}(x_t)} - \bar{F}_{t+1}(x_{t+1})].$$

$$\leqslant \sum_{t=1}^{T}[\bar{F}_{t+1}(x_t) - \bar{F}_{t+1}(x_{t+1})]$$

$$\leqslant \sum_{t=1}^{T}\left( \langle G_t, x_t - x_{t+1}\rangle - \frac{t\mu}{2}\|x_t - x_{t+1}\| \right).$$

Due to convexity, $G_t \in \partial \bar{F}_{t+1}(x_t) = \partial \bar{F}_t(x_t) \cap \partial f_t(x_t)$ and due to update operation $0 \in \partial \bar{F}_t(x_t) \cap \partial m_t(x_t)$. Thus, there exist $g_t \in \partial f_t(x_t)$ and $M_t \in \partial m_t(x_t)$ such that $G_t = g_t - M_t$, which completes the proof. $\square$

**Theorem 7.** *Under assumptions (A1)–(A4), the expected regret of OFTL with $M_t = \nabla f(x_{t-1}, \xi_{t-1})$ on surrogate loss functions $\ell_t$ defined in (4) is bounded as*

$$\mathbb{E}[R_T(u)] \leqslant \frac{1}{\mu}\sum_{t=1}^{T}\frac{1}{t}\left( 8\sigma_{\max}^2 + 4\mathbb{E}\left[ \sup_{x \in \mathcal{X}} \left\|\nabla F^t(x) - \nabla F^{t-1}(x)\right\|^2 \right] \right) + \frac{4D^2 L^2}{\mu}\log\left(1 + \frac{16L}{\mu}\right)$$

$$\leqslant \frac{1}{\mu}\left( 8\sigma_{\max}^2 + 4\Sigma_{\max}^2 \right)\log T + \frac{4D^2 L^2}{\mu}\log\left(1 + \frac{16L}{\mu}\right).$$

*Proof of Theorem 7.* Thanks to the strong convexity assumption **(A4)**,

$$F^t(x_t) - F^t(x) \leqslant \langle x_t - x, \nabla F^t(x_t)\rangle - \frac{\mu}{2}\|x - x_t\|^2.$$

Taking expectation and using the definition of $\ell_t$ gives

$$\mathbb{E}\left[F^t(x_t) - F^t(x)\right] \leqslant \mathbb{E}\left[\langle x_t - x, \nabla F^t(x_t)\rangle - \frac{\mu}{2}\|x - x_t\|^2\right]$$

$$= \mathbb{E}\left[\langle x_t - x, \nabla f(x_t, \xi_t)\rangle - \frac{\mu}{2}\|x - x_t\|^2\right]$$

$$= \mathbb{E}\left[\langle x_t - x, g_t\rangle - \frac{\mu}{2}\|x - x_t\|^2\right]$$

$$= \mathbb{E}\left[\ell_t(x_t) - \ell_t(x)\right].$$

Now each function $\ell_t$ is $\mu$-strongly convex, and $\nabla\ell_t(x_t) = g_t$. Thus we can apply Lemma 17

$$\sum_{t=1}^{T}\ell_t(x_t) - \ell_t(x) \leqslant \sum_{t=1}^{T}\left( \langle g_t - M_t, x_t - x_{t+1}\rangle - \frac{\mu t}{2}\|x_t - x_{t+1}\|^2 \right).$$

$$\leqslant \sum_{t=1}^{T}\left( \frac{1}{\mu t}\|g_t - M_t\|^2 + \left(\frac{\mu t}{4} - \frac{\mu t}{2}\right)\|x_t - x_{t+1}\|^2 \right).$$

where we used the inequality $\langle a, b\rangle \leqslant \frac{1}{2c}\|a\|^2 + \frac{c}{2}\|b\|^2$. Once again, keeping the negative norm term is crucial. Indeed, using the convexity of $x \mapsto \|x\|^2$ and the smoothness assumption on $\nabla F^t$, we get that for $t \geqslant 2$

$$\|g_t - g_{t-1}\|^2 \leqslant 4\|g_t - \nabla F^t(x_t)\|^2 + 4\|\nabla F^t(x_t) - \nabla F^t(x_{t-1})\|^2$$

$$+ 4\|\nabla F^t(x_{t-1}) - \nabla F^{t-1}(x_{t-1})\|^2 + 4\|\nabla F^{t-1}(x_{t-1}) - g_{t-1}\|^2$$

$$\leqslant 4\|g_t - \nabla F^t(x_t)\|^2 + 4L^2\|x_t - x_{t-1}\|^2$$

$$+ 4\|\nabla F^t(x_{t-1}) - \nabla F^{t-1}(x_{t-1})\|^2 + 4\|\nabla F^{t-1}(x_{t-1}) - g_{t-1}\|^2.$$

So that, upper bounding the first term $\ell_1(x_1) - \ell_1(u) \leqslant GD$ we get

$$\sum_{t=1}^{T} \ell_t(x_t) - \ell_t(u)$$

$$\leqslant \sum_{t=2}^{T} \frac{1}{\mu t}\left(4\|g_t - \nabla F^t(x_t)\|^2 + 4\|g_{t-1} - \nabla F^{t-1}(x_{t-1})\|^2 + 4\|\nabla F^t(x_{t-1}) - \nabla F^{t-1}(x_t)\|^2\right)$$

$$+ \sum_{t=1}^{T}\left(\frac{4L^2}{\mu(t+1)} - \frac{\mu t}{4}\right)\|x_t - x_{t+1}\|^2 + GD.$$

The indices can be simplified by noting that,

$$\sum_{t=2}^{T} \frac{4}{\mu t}\|g_{t-1} - \nabla F^{t-1}(x_{t-1})\|^2 \leqslant \sum_{t=2}^{T} \frac{4}{\mu(t-1)}\|g_{t-1} - \nabla F^{t-1}(x_{t-1})\|^2 \leqslant \sum_{t=1}^{T} \frac{4}{\mu t}\|g_t - \nabla F^t(x_t)\|^2.$$

To recover

$$\sum_{t=1}^{T} \ell_t(x_t) - \ell_t(u) \leqslant \sum_{t=1}^{T} \frac{8}{\mu t}\|g_t - \nabla F^t(x_t)\|^2 + \sum_{t=2}^{T} \frac{4}{\mu t}\|\nabla F^t(x_{t-1}) - \nabla F^{t-1}(x_t)\|^2$$

$$+ \sum_{t=1}^{T}\left(\frac{4L^2}{\mu t} - \frac{\mu t}{4}\right)\|x_t - x_{t+1}\|^2 + GD. \quad (13)$$

Define the condition number $\kappa = L/\mu$. Then, for $t \geqslant 16\kappa$, we have $\frac{4L^2}{\mu t} - \frac{\mu t}{4} \leqslant 0$. Therefore the second term can be bounded independently of $T$

$$\sum_{t=1}^{\lceil 16\kappa \rceil}\left(\frac{4L^2}{\mu t} - \frac{\mu t}{4}\right)D^2 \leqslant \frac{4L^2 D^2}{\mu} \sum_{t=1}^{\lceil 16\kappa \rceil} \frac{1}{t} \leqslant \frac{4L^2 D^2}{\mu} \log(1 + 16\kappa).$$

Combining all bounds, and incorporating the definition of $\sigma_{\max}$ and $\Sigma_{\max}$,

$$\mathbb{E}\left[R_T(u)\right] \leqslant \frac{1}{\mu}\left(8\sigma_{\max}^2 + 4\Sigma_{\max}^2\right)\log T + 4D^2 L\kappa \log(1 + 16\kappa) + GD. \qquad \square$$

## B.4   Proof of Theorem 8

**Theorem 8.** *For any learning algorithm, and for any pair of positive numbers $(\sigma, \Sigma)$ there exists a function $f : \mathcal{X} \times \Xi \to \mathbb{R}$ and a sequence of distributions satisfying assumptions (A1),(A2),(A3) and (A4) with $\sigma_{\max} \geqslant \sigma$ and $\Sigma_{\max} \geqslant \Sigma$ such that*

$$\mathbb{E}\left[R_T(u)\right] \geqslant \Omega\left(\frac{1}{\mu}\left(\sigma_{\max}^2 + \Sigma_{\max}^2\right)\log T\right).$$

*Proof of Theorem 8.* Let $\hat{\sigma}_{\max}, \hat{\Sigma}_{\max}$ be given parameters and set $G = \max(\hat{\sigma}_{\max}, \hat{\Sigma}_{\max}/2)$. We want to show that there exists sequence of distributions $\mathcal{D}_1, \ldots \mathcal{D}_T$ such that

1. $\sigma_{\max} = \hat{\sigma}_{\max}$ and $\Sigma_{\max} = \hat{\Sigma}_{\max}$.

2. $\mathbb{E}\left[R_T(u)\right] \geqslant c\frac{1}{\mu}(\Sigma_{\max}^2 + \sigma_{\max}^2)\log T$ for some constant $c > 0$.

3. $F_1, \ldots, F_T$ are $\mu$-strongly convex.

Consider the iterations up to $T - 3$. From Corollary 20 in [13] we obtain an $\Omega(\frac{1}{\mu}G^2 \log(T - 3))$ lower bound on the expected regret. Thus, there exist a realization $\xi_1, \ldots, \xi_{T-3}$ and corresponding $\mu$-strongly convex functions $f(\cdot, \xi_1), \ldots, f(\cdot, \xi_{T-3})$ such that with respect to this realization, $R_T(u) \geqslant \Omega(\frac{1}{\mu}G^2 \log(T - 3))$. We now let $\delta_1, \ldots \delta_{T-3}$ be the Dirac measure corresponding to this realization. Then, $\sigma_{\max} = 0$ and $\max_{t \in [T-3]} \sup_{x \in \mathcal{X}} \|\nabla F^t(x) - \nabla F^{t-1}(x)\| \leqslant 2G$. But we do not necessarily have that $\sigma_{\max} = \hat{\sigma}_{\max}$ and $\max_{t \in [T-3]} \sup_{x \in \mathcal{X}} \|\nabla F^t(x) - \nabla F^{t-1}(x)\| = \hat{\Sigma}_{\max}$. To guarantee this we want to choose $\mathcal{D}_{T-2}, \mathcal{D}_{T-1}, \mathcal{D}_T$, such that

1. $\sup_{x\in\mathcal{X}}\|\nabla F^{T-2}(x)-\nabla F^{T-1}(x)\|=\hat{\Sigma}_{\max}$ and $\|\nabla F^{T-i}(x)\|\leqslant G$ for $i=1,2$.

2. $\sigma_T=\hat{\sigma}_{\max}$ and $\|\nabla F^{T}(x)\|\leqslant G$.

To satisfy Condition 1, let $\mathcal{D}_{T-2},\mathcal{D}_{T-1}$ be Dirac measures, such that $\nabla F^{T-2}(x)=-\nabla F^{T-1}(x)$ and $\|\nabla F^{T-2}(x)\|=\frac{1}{2}\hat{\Sigma}_{\max}$. Then, by definition of $G$, we know that $\|\nabla F^{T-2}(x)\|\leqslant G$ and $\sup_{x\in\mathcal{X}}\|\nabla F^{T-2}(x)-\nabla F^{T-1}(x)\|=\hat{\Sigma}_{\max}$. Condition 2 can be satisfied by setting $\mathcal{D}_T$ to be any distribution with sufficient variance. This gives

$$\mathbb{E}\left[R_T(u)\right]\geqslant c\frac{1}{\mu}\left(\Sigma_{\max}^2+\sigma_{\max}^2\right)\log(T-3)-\mathbb{E}\left[\sum_{t=T-2}^{T}f(x_t,\xi_t)-f(u,\xi_t)\right].\qquad(14)$$

Now it remains to show that the last term is negligible. Indeed, from the upper bound, we know

$$\mathbb{E}\left[\sum_{t=T-2}^{T}[f(x_t,\xi_t)-f(u,\xi_t)]\right]\leqslant\frac{3}{(T-2)\mu}\left(\Sigma_{\max}^2+\sigma_{\max}^2\right).$$

Hence, for any $T\geqslant 10$, we get $\frac{3}{(T-2)\mu}\left(\Sigma_{\max}^2+\sigma_{\max}^2\right)\leqslant\frac{1}{2\mu}\left(\Sigma_{\max}^2+\sigma_{\max}^2\right)\log(T)$ which together with (14) completes the proof. $\qquad\square$

## C  Missing Proofs of Section 4

We first show the following general property of the variance for the ROM. This proposition will useful for showing the claims of this section.

**Proposition 18.** *For any $t\in[T]$, the variance of the ROM with respect to $\mathcal{D}_t$ satisfies*

$$\mathbb{E}_{\xi\sim\mathcal{D}_t}\left[\|\nabla f(x,\xi)-\nabla F^t(x)\|^2\right]\leqslant\frac{T}{T-t+1}\sigma_1^2,$$

*for any $x\in\mathcal{X}$.*

*Proof.* For any $x\in\mathcal{X}$, since $\nabla F^t(x)=\mathbb{E}_{\xi\sim\mathcal{D}_t}[\nabla f(x,\xi)]$, we have

$$\mathbb{E}_{\xi\sim\mathcal{D}_t}\left[\|\nabla f(x,\xi)-\nabla F^t(x)\|^2\right]\leqslant\mathbb{E}_{\xi\sim\mathcal{D}_t}\left[\|\nabla f(x,\xi)-\nabla F^1(x)\|^2\right].$$

Now, let $\mathcal{T}_t\subseteq[T]$ denote a subset of indices of gradients which remain to be selected in round $t$, and let $k_t\in\mathcal{T}_{t-1}\setminus\mathcal{T}_t$ be the index selected at round $t$.

For any $x\in\mathcal{X}$

$$\mathbb{E}_{\xi\sim\mathcal{D}_t}\left[\|\nabla f(x,\xi)-\nabla F^1(x)\|^2\right]=\frac{1}{T-t+1}\sum_{\xi\in\mathcal{T}_t}\|\nabla f(x,\xi)-\nabla F^1(x)\|^2$$

$$\leqslant\frac{1}{T-t+1}\sum_{\xi\in[n]}\|\nabla f(x,\xi)-\nabla F^1(x)\|^2\leqslant\frac{T}{T-t+1}\sigma_1^2,\qquad(15)$$

which is the claimed result. $\qquad\square$

### C.1  Proof of Lemma 11

Note that in any case, $\tilde{\sigma}_1\leqslant T\sigma_1$, and therefore $\log(\tilde{\sigma}_1/\sigma_1)\leqslant\log(T)$. Thus Lemma 11 directly yields $\sigma_{[1:T]}^{(2)}\leqslant\sigma_1 T\log(T)$. This means in particular that the rate of OFTRL in the ROM is never more than a factor $\sqrt{\log T}$ worse than the i.i.d. sampling with replacement rate of $\sigma_1\sqrt{T}$; the next bound can often be much tighter.

**Lemma 11.** *In the single-pass ROM, we have $\Sigma_{[1:T]}^{(2)}\leqslant 8G^2$ and $\bar{\sigma}_{[1:T]}^{(2)}\leqslant T\sigma_1^2\log(2e^2\tilde{\sigma}_1^2/\sigma_1^2)$.*

*Proof of Lemma 11.* Let us begin with the adversarial variation. We will show that deterministically (that is, for any order in which the losses are selected), for any $x \in \mathcal{X}$,

$$\left\| \nabla \mathrm{F}^t(x) - \nabla \mathrm{F}^{t-1}(x) \right\|^2 \leqslant \frac{4G^2}{(T-t+2)^2} \, .$$

With the same notation as in Proposition 18, recall that we denote by $\mathcal{T}_t \subseteq [T]$ the support of $\mathcal{D}_t$ and $k_t = \mathcal{T}_{t-1} \setminus \mathcal{T}_t$. We have $|\mathcal{T}_t| = T - t + 1$ and for any $x \in \mathcal{X}$,

$$\left\| \nabla \mathrm{F}^t(x) - \nabla \mathrm{F}^{t-1}(x) \right\|^2 = \left\| \frac{1}{T-t+1} \sum_{s \in \mathcal{T}_t} \nabla f_s(x) - \frac{1}{T-t+2} \sum_{s \in \mathcal{T}_{t-1}} \nabla f_s(x) \right\|^2$$

$$= \left\| \frac{1}{(T-t+1)(T-t+2)} \sum_{s \in \mathcal{T}_t} \nabla f_s(x) - \frac{1}{T-t+2} \nabla f_{k_t}(x) \right\|^2$$

$$\leqslant \frac{2}{(T-t+2)^2} \left\| \frac{1}{(T-t+1)} \sum_{s \in \mathcal{T}_t} \nabla f_s(x) \right\|^2 + \frac{2}{(T-t+2)^2} \left\| \nabla f_{k_t}(x) \right\|^2 \, .$$

Thus, after maximising over $x \in \mathcal{X}$, and taking expectations (note that the inequality holds almost surely) and summing over rounds $t \in [T]$,

$$\Sigma^{(2)}_{[1:T]} = \mathbb{E}\left[ \sum_{t=1}^{T} \sup_{x \in \mathcal{X}} \left\| \nabla \mathrm{F}_t(x) - \nabla \mathrm{F}_{t-1}(x) \right\|^2 \right] \leqslant \sum_{t=1}^{T} \frac{4G^2}{(T-t+2)^2} \leqslant 8G^2.$$

**Variance.** From Proposition 18, we know that

$$\sigma_t^2 \leqslant \frac{T}{T-t+1} \sigma_1^2$$

Moreover, one can see that

$$\mathbb{E}[\sigma_t^2] \leqslant \mathbb{E}\left[ \max_{x \in \mathcal{X}} \mathbb{E}_{\xi \sim \mathcal{D}_t} \left[ \|\nabla f(x, \xi) - \nabla F^1(x)\|^2 \right] \right]$$

$$\leqslant \mathbb{E}\left[ \mathbb{E}_{\xi \sim \mathcal{D}_t} \left[ \max_{x \in \mathcal{X}} \|\nabla f(x, \xi) - \nabla F^1(x)\|^2 \right] \right] = \tilde{\sigma}_1^2. \quad (16)$$

Let us introduce a threshold time step $\tau \in [T]$, of which we will set the value later. We upper bound $\mathbb{E}[\sigma_t^2]$ by (15) for the rounds before $\tau$ and by (16) for the other rounds:

$$\mathbb{E}\left[ \sum_{t=1}^{T} \sigma_t^2 \right] \leqslant \mathbb{E}\left[ \sum_{t=1}^{\tau} \sigma_t^2 \right] + \mathbb{E}\left[ \sum_{t=\tau+1}^{T} \sigma_t^2 \right] \leqslant \sum_{t=1}^{\tau} \frac{T}{T-t+1} \sigma_1^2 + (T-\tau)\tilde{\sigma}_1^2$$

Now using standard bounds on the harmonic series,

$$\sum_{t=1}^{\tau} \frac{1}{T-t+1} = \sum_{n=T-\tau+1}^{T} \frac{1}{n} \leqslant 1 + \log \frac{T}{T-\tau+1} \, .$$

Therefore for any $\tau \in [T-1]$, we get

$$\mathbb{E}\left[ \sum_{t=1}^{T} \sigma_t^2 \right] \leqslant T\sigma_1^2 \left( 1 + \log \frac{T}{T-\tau+1} \right) + (T-\tau)\tilde{\sigma}_1^2 \, . \quad (17)$$

We now conclude by setting the appropriate value for $\tau$. If $T\sigma_1^2/\tilde{\sigma}_1^2 \leqslant 2$, then $\log T \leqslant \log(2\tilde{\sigma}_1^2/\sigma_1^2)$, and taking $\tau = T$ gives a bound of $T\sigma_1^2(1 + \log T) \leqslant T\sigma_1^2(1 + \log(2\tilde{\sigma}_1^2/\sigma_1^2))$, which is (better than) the claimed result.

Otherwise, we take $\tau = T - \lfloor T\sigma_1^2/\tilde{\sigma}_1^2 \rfloor$, then

$$(T-\tau)\tilde{\sigma}_1^2 = \lfloor T\sigma_1^2/\tilde{\sigma}_1^2 \rfloor \tilde{\sigma}_1^2 \leqslant T\sigma_1^2 \, ,$$

and the argument of the logarithm can be bounded as

$$\frac{T}{T - \tau + 1} \leqslant \frac{T}{\lfloor T\sigma_1^2/\tilde{\sigma}_1^2 \rfloor} \leqslant \frac{1}{\sigma_1^2/\tilde{\sigma}_1^2 - 1/T} \leqslant \frac{2\tilde{\sigma}_1^2}{\sigma_1^2}.$$

where we used the fact that $T\sigma_1^2/\tilde{\sigma}_1^2 > 2$. This yields the final bound

$$\mathbb{E}\left[\sum_{t=1}^{T} \sigma_t^2\right] \leqslant T\sigma_1^2\left(1 + \log\frac{2\tilde{\sigma}_1^2}{\sigma_1^2}\right) + T\sigma_1^2 \leqslant T\sigma_1^2\log\left(\frac{2e^2\tilde{\sigma}_1^2}{\sigma_1^2}\right). \qquad \square$$

## C.2    Proof of Corollary 12

**Corollary 12.** *Under the same assumption as in Theorem 7, the expected regret of the ROM is bounded by*

$$\mathbb{E}\left[R_T(u)\right] \leqslant O\left(\frac{\sigma_1^2}{\mu}\log T + \frac{G^2}{\mu} + LD^2\kappa\log\kappa\right).$$

*For multi-pass ROM with $P$ passes, we obtain*

$$\mathbb{E}\left[R_T(u)\right] \leqslant O\left(\frac{\sigma_1^2}{\mu}\log T + \frac{G^2}{\mu} + \frac{G^2\log P}{n\mu} + LD^2\kappa\log\kappa\right).$$

*Proof of Corollary 12.* **Single-pass ROM:** From Theorem 7 we obtain (c.f. (13))

$$\mathbb{E}\left[R_T(u)\right] \leqslant \mathbb{E}\left[\sum_{t=1}^{T}\frac{8}{\mu t}\|g_t - \nabla F^t(x_t)\|^2 + \sum_{t=2}^{T}\frac{4}{\mu t}\|\nabla F^t(x_{t-1}) - \nabla F^{t-1}(x_t)\|^2\right]$$
$$+ GD + \frac{4L^2D^2}{\mu}\log(1 + 16\kappa).$$

By Lemma 11, we have

$$\mathbb{E}\left[\sum_{t=2}^{T}\frac{4}{\mu t}\|\nabla F^t(x_{t-1}) - \nabla F^{t-1}(x_t)\|^2\right] \leqslant 8G^2.$$

Furthermore, recall that by Proposition 18 $\mathbb{E}[\sigma_t^2] \leqslant T/(T - t + 1)\sigma_1^2$.

$$\mathbb{E}\left[\sum_{t=1}^{T}\frac{8}{\mu t}\|g_t - \nabla F^t(x_t)\|^2\right] \leqslant \frac{8}{\mu}\sum_{t=1}^{T}\frac{T}{t(T - t + 1)}\sigma_1^2 \leqslant \frac{8\sigma_1^2}{\mu}(2 + 2\log(T)).$$

Indeed, using a standard bound on the harmonic series,

$$\sum_{t=1}^{T}\frac{T}{t(T - t + 1)} = \sum_{t=1}^{T}\frac{T - t + 1 + t - 1}{t(T - t + 1)} \leqslant \sum_{t=1}^{T}\frac{1}{t} + \frac{1}{T - t + 1} \leqslant 2 + 2\log T.$$

Combining these bounds gives the first part of the corollary.

**Multi-pass ROM:** The critical term to upper bound, is the differences of the means whenever a pass ends and a new pass starts. Thus, for $P \in \mathbb{N}$ passes, we need to control $\sup_{x\in\mathcal{X}}\left\|\nabla F^t(x) - \nabla F^{t-1}(x)\right\|^2$ for $t = jn + 1$, with $j \in [P]$.

Inside the $i$-th pass, for $k \in [n]$ we bound the $k$-th variation by

$$\sup_{x\in\mathcal{X}}\left\|\nabla F^k(x) - \nabla F^{k-1}(x)\right\|^2 \leqslant \frac{4G^2}{(n - k + 2)^2},$$

and we bound it by $G^2$ between the passes, so that

$$
\frac{4}{\mu} \sum_{t=1}^{T} \frac{1}{t} \sup_{x \in \mathcal{X}} \left\| \nabla F^t(x) - \nabla F^{t-1}(x) \right\|^2
$$

$$
\leqslant \frac{4}{\mu} \sum_{i=1}^{P} \sum_{k=1}^{n} \frac{1}{(i-1)n+k} \sup_{x \in \mathcal{X}} \left\| \nabla F^k(x) - \nabla F^{k-1}(x) \right\|^2 + \frac{4}{\mu} \sum_{i=1}^{P} \frac{1}{in} \sup_{x \in \mathcal{X}} \left\| \nabla F^1(x) - \nabla F^n(x) \right\|^2
$$

$$
\leqslant \frac{4}{\mu} \sum_{i=1}^{P} \sum_{k=1}^{n} \frac{1}{(i-1)n+k} \frac{2G^2}{(n-k+2)^2} + \frac{4}{\mu} \sum_{i=1}^{P} \frac{1}{in} \sup_{x \in \mathcal{X}} \left\| \nabla F^1(x) - \nabla F^n(x) \right\|^2
$$

$$
\leqslant \frac{16G^2}{\mu} \left( 1 + 2 \frac{\log P}{n} \right) . \qquad \square
$$

## D  Batch-to-online Conversion

Consider the stochastic optimization problem $\min_{x \in \mathcal{X}} \mathbb{E}_{\xi \sim \mathcal{D}} [f(x, \xi)]$ and let $x^*$ denote a minimiser for this problem. Further, let $\mathcal{A}$ be any first order stochastic optimization method with convergence guarantee $\mathbb{E}_{\xi \sim \mathcal{D}} [f(x_t, \xi) - f(x^*, \xi)] \leqslant c(t)$. As input $\mathcal{A}$ takes an initial iterate $x_1$ and a sequence of i.i.d. samples $\{f(\cdot, \xi_s)\}_{s \in [t]}$. We let $\mathcal{A}(x_1, \{f(\cdot, \xi_s)\}_{s \in [t]})$ denote the output $x_{t+1}$ of the stochastic optimization algorithm with respect to the given input. Now consider an OCO with $f(\cdot, \xi_1), \ldots f(\cdot, \xi_T)$ and $\xi_1, \ldots \xi_T$ are sampled i.i.d. from a distribution.

> **Input:** Stochastic first order method $\mathcal{A}$
> **for** $t = 1, 2, \ldots T$ **do**
>     play $x_t$ and suffer loss $f(x_t, \xi_t)$;
>     restart $\mathcal{A}$ and set $x_{t+1} = \mathcal{A}(x_1, \{f(\cdot, \xi_s)\}_{s \in [t]})$
> **end**

**Algorithm 1:** Batch-to-online

This batch-to-online conversion trivially achieves $\sum_{t=1}^{T} c(t)$ expected regret. However, with this conversion, some aspects of the stochastic convergence bound are lost. Consider for instance a convergence rate $c(t) = O(LD^2/t + D\sigma/\sqrt{t})$, from the the first-order stochastic approximation method in [7] and the accelerated version $c(t) = O(LD^2/t^2 + D\sigma/\sqrt{t})$ [8, 17]. In both cases, the functions are assumed to satisfy **(A1)** -**(A3)** . Batch-to-online conversion yields

$$
\mathbb{E}[R_T(u)] \leqslant O(LD^2 \log T + D\sigma \sqrt{T}) \qquad \text{and} \qquad \mathbb{E}[R_T(u)] \leqslant O(LD^2 + D\sigma \sqrt{T}).
$$

The benefits of acceleration can be seen in the lower-order terms. Now using standard online-to-batch [3] conversion gives the convergence bounds

$$
\mathbb{E}[f(x_T, \xi) - f(x^*, \xi)] \leqslant O\left( \frac{D\sigma}{\sqrt{T}} + LD^2 \frac{\log T}{T} \right) \text{ and } \mathbb{E}[f(x_T, \xi) - f(x^*, \xi)] \leqslant O\left( \frac{D\sigma}{\sqrt{T}} + \frac{LD^2}{T} \right) ,
$$

In the case of accelerated stochastic approximation, the benefits of acceleration are inevitably lost through batch-to-online and online-to-batch conversion.

## E  Additional Examples for Intermediate Cases

We provide regret bounds for intermediate cases not discussed in the main body of the paper, namely the cases when the adversary selects slowly shifting distributions and when the adversary switches rarely between distributions.

**Distribution shift:**     In this example, the SEA picks $\mathcal{D}_t$ and $\mathcal{D}_{t-1}$, such that $\nabla F^t(x)$ is close to the mean of the previous distribution gradient $\nabla F^{t-1}(x)$. We shall consider two kinds of distribution shifts. Firstly, when the means are close on average, that is, when $(1/T) \sum_{t=1}^{T} \sup_{x \in \mathcal{X}} \|\nabla F^t(x) - \nabla F^{t-1}(x)\|^2 \leqslant \varepsilon$, secondly, when this holds for each iteration

$t$, i.e., $\sup_{x \in \mathcal{X}} \left\| \nabla F^t(x) - \nabla F^{t-1}(x) \right\|^2 \leqslant \varepsilon$. We refer to the former as the *average distribution shift* case, and to the latter as the *bounded distribution shift* case.

For strongly convex functions, Theorem 7 directly yields the regret bound

$$\mathbb{E}\left[R_T(u)\right] \leqslant O\left(\frac{1}{\mu}(\sigma_{\max}^2 + \varepsilon)\log T + D^2 L\kappa \log \kappa\right).$$

For the considerably weaker assumption of an average distribution shift, we have

$$\sum_{t=1}^{T} \frac{1}{t\mu} \sup_{x \in \mathcal{X}} \left\| \nabla F^t(x) - \nabla F^{t-1}(x) \right\|^2 \leqslant \Sigma_{[1:T]}^{(2)} \sqrt{\sum_{t=1}^{T} \frac{1}{t^2 \mu^2}} \leqslant \frac{4}{\mu} T\varepsilon.$$

To obtain the first inequality, we have used the Cauchy-Schwarz inequality together with the fact that $\sqrt{a+b} \leqslant \sqrt{a} + \sqrt{b}$, and the second inequality follows directly from the definition of the averaged distribution shift. Now suppose $\varepsilon \leqslant 1/T$, then we obtain the following regret bound in case of average distribution shift.

$$\mathbb{E}\left[R_T(u)\right] \leqslant O\left(\frac{\sigma_{\max}^2}{\mu} \log T + \frac{1}{\mu} + D^2 L\kappa \log \kappa\right).$$

Since $\Sigma_{[1:T]}^{(2)} \leqslant T\varepsilon$ for the average distribution shift, for convex and smooth functions, Theorem 5 entails that

$$\mathbb{E}\left[R_T(u)\right] \leqslant O\left(D(\sigma_{\max} + \sqrt{\varepsilon})\sqrt{T} + DG + LD^2\right).$$

**Distribution switch:** SEA switches $c$ times between distributions $\mathcal{D}_1, \ldots, \mathcal{D}_c \in \mathfrak{D}$. These switches can happen at any round and the learner does not know when a switch occurs. In this case, we can upper bound $\Sigma_{[1:T]}^{(2)} \leqslant \Sigma_{\max}^2 c$. Thus, for strongly convex functions Theorem 7 directly yields

$$\mathbb{E}\left[R_T(u)\right] \leqslant O\left(\frac{1}{\mu}\left(\sigma_{\max}^2 \log T + \Sigma_{\max}^2 \log c\right) + D^2 L\kappa \log \kappa\right).$$

And for convex smooth functions Theorem 5 gives

$$\mathbb{E}\left[R_T(u)\right] \leqslant O\left(D\sigma_{\max}\sqrt{T} + D\Sigma_{\max}\sqrt{c} + DG + LD^2\right).$$