# OpenReview forum: "Between Stochastic and Adversarial Online Convex Optimization: Improved Regret Bounds via Smoothness"
_NeurIPS.cc/2022/Conference — NeurIPS 2022 Accept_

### Official Review · Reviewer_KxnX · 2022-07-06

**Rating:** 6
**Confidence:** 4
**Soundness:** 4 excellent
**Presentation:** 3 good
**Contribution:** 3 good

**Summary:**

Examines a range of stochastic optimization settings from IID to Adversarial.
Provides regret bounds for optimistic FTRL in terms of a combination "within round variances" and "between round variances", the former representing the stochastic part of the gradients and the latter representing the adversarial part, recovering good known bounds for both of the pure iid and pure adversarial cases if they prevail, and interpolating "naturally" between these cases.
Both in the convex and strongly convex cases.
Lower bounds that match time-dependent term.






**Questions:**

- unclear why lag-1 variability of the gradient is the right notion of "adversarial". if the player has access to f( . , \xi_t) for all past t, for example, then one could predict with the average of the history instead of the lag-1 predictor. does this lead to a different upper/lower bound? is there a way in which not using all the history is better? I imagine one could do better using the full history.

**Limitations:**

some discussion of other settings between iid and adversarial that would not be covered, or would not achieve their respective optimal bounds using these methods would be useful to provide context to the reader.

**Strengths And Weaknesses:**

Strengths:
- Interesting, novel setting.
- Results are intuitive and interpretable in the context of related work.
- "best of both worlds" gets optimal rates in multiple settings without knowledge of the setting to tune the algorithm.
- the examples provided on pgs 4/5 make it clear that this these results are useful and flexible as they can be applied to a wide range of settings that are of practical relevance.
- proofs seem to me to be correct.
- topic of "between iid and adversarial" is timely and interesting

Weaknesses:
- The proof techniques are not novel. The novelty of the results is essentially due to the setting that was introduced and application of the corresponding definitions the authors introduced. However, this is not a big detractor, in my opinion, since the results themselves are interesting regardless.

---

> ### Author Response · Authors · 2022-08-01
> **Answer to 'Weaknesses' section and 'Question'**
>
> We answer both the question and the remark on limitations of the setting simultaneously.
>
> About the use of the last gradient vs. the average (i.e.,  $M_t = g_{t-1}$ vs. $M_t = (g_1 + \dots + g_{t-1})/(t-1)$). Informally, $M_t$ should be an estimate of the next gradient $g_t$ (unobserved yet).  In the i.i.d. case, using the last gradient would be a better estimate of $g_t$ than the average of the observed gradients.
>
> Our matching lower bounds show that there are instances where lag-1 captures the right adversarial behaviour. However, and this relates to the remarks about limitations and lag-1 bounds, there are cases where other measures of regularity would be more relevant. One such example would be an adversary oscillating between two distributions. Interestingly, using $M_t = g_{t-2}$ would then recover optimal bounds.
>
> An interesting direction that exceeds the scope of our work would be to investigate what can be achieved when
> using linear combinations of past gradients as $M_t$. Note that using an off-the-shelf aggregating algorithm to learn the best linear combination would give an overhead scaling (at least) with the largest gradients again, leading to new technical challenges.
>
> We thank you for this interesting question and will add a discussion about it. (We already had a small discussion in an earlier arXiv version, but removed it due to the page constraint.)

---

> > ### Comment · Reviewer_KxnX · 2022-08-09
> > **response to response**
> >
> > I don't really follow why the last gradient would be better than the average in the IID case. Predicting with the true mean instead of instead of an independent draw of the same distribution halves the variance, and the empirical mean will converge to the true mean.
> >
> > I think the rest of the response answers my question adequately though.

---

> > > ### Author Response · Authors · 2022-08-09
> > > **response to response to response**
> > >
> > > You are right, the statement 'In the i.i.d. case, using the last gradient would be a better estimate of  $g_t$ than the average of the observed gradients.’ is a bit vague. Please let us clarify this:
> > > You are right that averaging more past gradients would reduce the variance of the estimator, potentially gaining a factor of 2. However, for $s < t$, $g_s$ is also a biased estimator of the mean of $g_t$, with the bias depending on the distance between $x_s$ and $x_t$, multiplied by the smoothness constant $L$. For $s = t-1$, $x_s$ and $x_t$ will be close and this bias is under control, but for $s \ll t$, the bias could be large when $L > 0$, so averaging the early gradients can potentially incur (too) large bias. This issue vanishes for linear losses, for which $L=0$ and the gradients $g_s$ do not depend on the iterates $x_s$, so for linear losses it is possible to average all past gradients after all.

---

### Official Review · Reviewer_AbNA · 2022-07-08

**Rating:** 7
**Confidence:** 3
**Soundness:** 4 excellent
**Presentation:** 4 excellent
**Contribution:** 3 good

**Summary:**

The paper gives tight results on optimization in a regime between stochastic and adversarial data, interpolating known results in these extremes. Using smoothness of the expected loss, the paper establishes convergence rates depending on stochastic variance and adversarial variation of the gradients via a unifying analysis of OFTRL. The authors discuss applications of the result in the intermediate regime, including adversarial corruption and in the random order model.


**Questions:**

- The regret in the corruption model is against the corrupted data. Could the authors comment on scenarios where this makes sense?
- Could the authors give some examples of problems where the results for random order models can be applied?


**Limitations:**

Yes.

**Strengths And Weaknesses:**

Strengths:
- The paper is very clearly written and discusses relevant works.
- In terms of technical novelty, this work proposes a new analysis of optimistic FTRL that takes into account a negative term not explicitly stated in classical analysis.
- The regret analysis is unified for the adversarial and stochastic cases, and the regret bounds are not only tight up to constants, but also match known results in the extreme cases.

Weaknesses:
- See questions.

Minor Typos:
- Page 7, $\Sigma_{[1:T]}^2 \le G\sqrt{2T}$, should there be a square on the lhs?
- Appendix Page 16, Eq. (9), should it be $\frac{6D^2G^2}{\nu}$?

---

> ### Author Response · Authors · 2022-08-01
> **Answers to the questions**
>
> We thank Reviewer AbNA for the interesting questions and for pointing us to several typos!
>
> **Q1:**
> Thank you for this important remark. Indeed, as noticed by [Amir et al.'20 (Observation 4)] and [Ito'21 (Remark 1)] in the experts setting, a bound against the corrupted losses implies one against the uncorrupted losses: in our notation and with our algorithm,
> \[
> \mathbb E\bigg[\sum_{t=1}^T F(w_t)- F(w^\star)\bigg] \leq \mathbb E[R_T] + 2D \mathbb E \bigg[\sum_{t=1}^T \|\nabla c_t(w_t)\| \bigg] \leq \mathcal O (D \sigma \sqrt T + D\sqrt{GC} + 2DC) \, ,
> \]
> that is, the sum of the stochastic rate and of a term linear in the cost for corruptions.
>
> As a side note regarding terminology: we followed the convention from the experts/bandits literature [Ito'21, Zimmert and Seldin'14] of calling $(c_t)$ corruptions. We realize that this may imply that the $(c_t)$ are not part of the 'true losses'. However, depending on modelling choices, the $(c_t)$ can be seen as 'true' deviations from an i.i.d. model, in which case the relevant quantity to bound is indeed $R_T$.
>
> We will add these comments to the final version.
>
> *References*
>
> Amir, I., Attias, I., Koren, T., Mansour, Y., and Livni, R. (2020). Prediction with corrupted expert advice. Advances in Neural Information Processing Systems, 33, 14315-14325.
>
> Ito, S. (2021). On optimal robustness to adversarial corruption in online decision problems. Advances in Neural Information Processing Systems, 34, 7409-7420.
>
> Zimmert, J., and Seldin, Y. (2021). Tsallis-INF: An Optimal Algorithm for Stochastic and Adversarial Bandits. J. Mach. Learn. Res., 22(28), 1-49.
>
>
> **Q2.**
> Note that shuffling datasets before optimization is common practice in machine learning, and that the ROM is relevant in modelling this practice.
> As for applications, our results provide data-dependent refinements of the worst-case bounds of [Sherman et al. '21], albeit partial refinements, since we require individual convexity of losses. We hope to be able generalize our data-dependent bounds to more general settings in future work.
>
> For a more practical application consider the following two examples:
>  1. Online-PCA problem as considered in Garber et al.
>  2. Profit maximization as considered by Gupta et al. This problem attracted some interest in the theoretical computer science community. Our results are directly applicable to this problem and add another insight in addition to the known competitive ratios.
>
> *References*
>
> Sherman, U., Koren, T., and Mansour, Y. (2021). Optimal rates for random order online optimization. Advances in Neural Information Processing Systems, 34, 2097-2108.
>
> Garber, D., Korcia, G., Levy, K.Y. (2020). Online Convex Optimization in the Random Order Model. Proceedings of the 37th International Conference on Machine Learning, Online, PMLR 119.
>
> Gupta, A., Mehta, R., Molinaro, M. (2018) Maximizing Profit with Convex Costs in the Random-order Model. 45th International Colloquium on Automata, Languages, and Programming (ICALP 2018)

---

### Official Review · Reviewer_nGYM · 2022-07-09

**Rating:** 6
**Confidence:** 2
**Soundness:** 3 good
**Presentation:** 2 fair
**Contribution:** 3 good

**Summary:**

This work studies regret bounds for online convex optimization that interpolates between stochastic i.i.d. and fully adversarial losses. The author leverages the smoothness of the expected loss to reduce the dependence of the regret bound on the maximal gradient norm to the cumulative stochastic variance and the adversarial variation.

**Questions:**

1. For theorem 5, can you elaborate more on why you specifically choose $\nu=LD^2+DG^2$ and $\nu=2DG$ after deriving eq(3)?

2. Why in theorem 6 and theorem 8 we need $\bar\sigma_T\geq\sigma$ and $\bar\Sigma_T\geq \Sigma$ since there’s no $\sigma$ and $\Sigma$ show up in the bound?

3. It’d be helpful to also provide the lower bound for adversarially corrupted i.i.d. model to see whether the dependence on C is tight. Also, does it natural to bound the sum of perturbation by $C$, instead of per sample perturbation $\max_{t\in T}\max_{x\in X}\|\nabla c_t(x)\|\leq C$? Since the latter case might provide an $O(\sqrt{T})$ bound. It’d also be helpful if we can have a lower bound for Random Order Models.

4. Some typo: in the appendix proof of theorem 5, I think eq(9) the constant before $D^2G^2/\mu$ should be 6 instead of 4. line 569-571 some of the summation is not consistent, which should be from 1 to $T-1$. It seems a missing constant in front of $G$ in the equation after line 569.



##############After reading the rebuttal################
I thank the author for the detailed response to my question. Most of my concerns have been addressed. I therefore raise my score to 6 for now.

**Limitations:**

Yes.

**Strengths And Weaknesses:**

Strengths:
1. This work provides upper bound and tight lower bound up to additive constants of regret bounds for (strongly) convex smooth function.

2. Such results can be extended to the intermediate case between stochastic iid to fully adversary.


Weaknesses:
1. The writing needs to be improved. It’s hard to find the definition of some math parameters. For example, there’s no context regarding $D, \kappa, \sigma_{max}, \Sigma_{max}$ when introducing the bound in line 47-49. When comparing with the literature, it’d be helpful to list the concrete differences. For example, line 93 mentioned the main theorem under stronger assumptions than previous literature without mentioning what’s the assumption in this work versus the literature. In Remark 4 line 212, the author does not give any formal definition of $g_t$ and $M_t$.

2. The author introduces the smoothness of the expected loss function as one additional assumption besides bounded gradient. To the best of my knowledge, there exists expected-smoothness assumption [1] that also try to leverage the smoothness of the function, yet can replace the bounded gradient and bounded variance assumption. I’m wondering whether the author considers other methods to try to eliminate some of the assumptions.


[1] Gower, Robert Mansel, et al. "SGD: General analysis and improved rates." International Conference on Machine Learning. PMLR, 2019.

---

> ### Author Response · Authors · 2022-08-01
> **Answer to the comments in the 'Weaknesses' section**
>
>
>
>  1.:
> In the introduction, we aim to give a high-level description of our results without going into technical detail; formal definitions are provided in the 'Setting' section, and precise discussion on assumptions in the appropriate sections. Regarding the Random Order Model (line 93) we explain precisely the difference with the literature on line 329.
> We agree that some aspects can be improved and thank the reviewer for the detailed comments:
>
> - we will add the missing high-level descriptions of parameters in the introduction (line 47-49), and point to their formal definitions.
>
> - we will add missing definitions of $g_t$ and $M_t$ and point to them in Remark 4.
>
> 2.:
> Thank you for pointing us to this segment of the literature. Please note that the paper [Gower et al.'19] considers stochastic optimization of a finite sum problem (with arbitrary sampling). Their results rely crucially on the finite sum structure, which on the one hand is making them incomparable to ours in general and on the other hand is fundamentally different from the online setting that motivates our work.
>
> Note in particular that the randomness in this setting comes from the sampling among the elements of the finite sum, implying a very specific structure on the noise and variance.
> Furthermore, none of their key assumptions, i.e.,  the finite noise assumption for the minimizer (Assumption 2.3) nor the 'expected smoothness' assumption (Assumption 2.1), have a natural analogue in the sequential setting, since they involve the minimizer of the finite sum problem $x^\star$.
>
> While we agree that looking for weaker assumptions is worthwhile, we do not think this particular line of work is close enough to our setting to be fruitfully applied.
>
>
> *References:*
>
> Gower, R. M., Loizou, N., Qian, X., Sailanbayev, A., Shulgin, E., Richtarik, P. (2019)  SGD: General analysis and improved rates.  International Conference on Machine Learning. PMLR, 2019.

---

> ### Author Response · Authors · 2022-08-01
> **Answers to the questions**
>
> **Q1:**
> There is a typo, $DG^2$ should $DG$. The value of $\nu$ is chosen to approximately minimize $\nu +  \frac{L^2D^4 + G^2D^2}{\nu}$.
>
> **Q2:**
> Lower bounding the values of $(\bar \sigma_T , \bar \Sigma_T)$ is crucial to state a meaningful lower bound. Otherwise, a constant sequence of losses with $(\bar \sigma_T , \bar \Sigma_T)= (0, 0)$ would yield a valid yet vacuous lower bound.
> Then, writing the equation in the lower bound with $(\bar \sigma_T , \bar \Sigma_T)$ or with $(\sigma, \Sigma)$ is just a matter of style and both would have essentially the same meaning. We chose the former to match visually the upper bounds.
>
> **Q3:**
>  We are not sure if we understood your conjecture about a potential $O(\sqrt{T})$ bound correctly. The main contribution of our work is a data-dependent refinement of the regret bounds. Note that by our assumptions $D,\sigma, G, C$ are independent of $T$ thus can be incorporated as constants in the asymptotic notation. Then Cor. 9  directly implies a $O(\sqrt{T})$ regret bound.  Note that this bound is much weaker than what we provide. Furthermore, this bound can be obtained by well known results (e.g. Zinkevich '03) and is therefore not a meaningful application example for our refined results. If we misunderstood your comment, please let us know and clarify during the discussion period.
>
> *Regarding different corruption measures.* It is indeed possible to use another norm on the corruptions, and choosing a measure is a modelling question. Note however that the existing literature on corruptions in experts [Amir et al.'20, Ito '21] and bandits [Zimmert and Seldin '21] also use the cumulative norm of the corruptions.
>
> To avoid confusion in the subsequent discussion, denote the constant for the cumulative bound by $C_c = \sum_{t=1}^T \max_{x \in X} \|\nabla c_t(x)\| $ and for the uniform bounded corruption by $C_u = \max_{t\in[T]} \max{x \in X} \|\nabla c_t(x)\|$. Note that $C_c \leq T C_u$.
> Observe that $\Sigma_{[1:T]}^{(2)} \leq 2TG \max_{t \in [T]} \max_{x \in X} \|\nabla c_t(x)\| = 2GC_uT$, thus our bounds also apply to the uniformly bounded corruption case you are suggesting. Indeed, Thm. 5 directly yields a regret bound of $O(D(\sigma + \sqrt{G C_u})\sqrt{T} )$.
> (See also the next paragraphs for a comment on the tightness of regret bounds obtained with this approximation.)
>
> *Regarding optimality in the applications.* The goal of the applications is merely to illustrate the relevance and flexibility of our framework and algorithms: a more in-depth investigation of both settings would be interesting but it is out of the scope of our work. Nevertheless, we agree with the reviewer that some extra comments on optimality would be a nice addition.
>
> *a. Corruptions.*
>  The constructions in our lower bound Theorem~6 can also provide lower bounds for the corruption model, showing that our rates are unimprovable in the worst case for both corruption models.
>
> Indeed, for any value of $C$ and for any algorithm, consider a sequence $c_t$ with gradients of size $G = C$ coming from a lower bound in OCO. This can be seen as corruptions applied to identically null stochastic data. The lower bound for standard OCO applies to the regret with $R_T \geq DG\sqrt{T} = D \sqrt{G} \sqrt{CT}$. For $C = C_u$ this matches our upper bound for the uniform bounded corruption model. For the cumulative model note that the worst-case sequences of losses can be chosen to be linear with norm equal to $G$ (see, e.g. [Hazan '16, Thm. 3.2]). In this case, we have $C_c = \sum \|\nabla c_t\| = T \max \|\nabla c_t\| = T C_u$, thus implying a lower bound for the cumulative model.
>
> Conversely, in the absence of perturbations, $c_t \equiv 0$, then applying Theorem~6 yields $R_T \geq D \sigma \sqrt T$.
>
> *b. ROM.*
> A matching lower bound for the ROM in the convex case is more challenging. We conjecture that the logarithmic factors could be removed with a more detailed analysis. Also note that this is related to the question of matching upper and lower bounds for random reshuffle SGD which is, to the best of our knowledge, not resolved (see end of section 3.2 in [Mishchenko et al.'20]).
>
>
> **Q4:** We thank you for pointing us to the typos and we will correct them.
>
>
> *References*
>
> Hazan, E. (2016). Introduction to online convex optimization. Foundations and Trends® in Optimization, 2(3-4), 157-325.
>
> Mishchenko, K., Khaled, A., Richtarik, P. (2020) Random Reshuffling: Simple Analysis with Vast Improvements. 34th Conference on Neural Information Processing Systems (NeurIPS 2020), Vancouver, Canada.
>
> Zinkevich, M. (2003) Online convex programming and generalized infinitesimal gradient ascent. Proceedings of the Twentieth International Conference on International Conference on Machine Learning, ICML’03, pages 928–935. AAAI Press, 2003.

---

### Author Response · Authors · 2022-08-01
**Discussion**

We thank all reviewers for their careful reading of the paper and their detailed remarks. We note that all reviewers appreciated the novelty and relevance of the setting introduced. We hope we have addressed their concerns and will stay available during the discussion period to clarify remaining points.

---

### Meta-Review · Area_Chair_oGdV · 2022-08-31

**Recommendation:** Accept
**Confidence:** Certain

**Metareview:**

While the reviewers had some concerns that were raised in the reviews, overall the reviewers seem to be leaning towards an accept. The setting/motivation is a nice one and the result is clean. I agree with the reviewers in that the paper should be accepted for publication.

**Award:**

No

---

### Decision · Program_Chairs · 2022-09-14

Accept